# nature | methods

ANALYSIS

# Analysis of the Human Protein Atlas Weakly Supervised Single-Cell Classification competition

Trang Le[1,2], Casper F. Winsnes[1], Ulrika Axelsson[1], Hao Xu[1], Jayasankar Mohanakrishnan Kaimal[1], Diana Mahdessian[1], Shubin Dai[3], Ilya S. Makarov[4], Vladislav Ostankovich[5], Yang Xu[6], Eric Benhamou[7], Christof Henkel[8], Roman A. Solovyev[9], Nikola Banić[10], Vito Bošnjak[11], Ana Bošnjak[12], Andrija Miličević[13], Wei Ouyang[1] and Emma Lundberg[1,2,14,15] ✉

While spatial proteomics by fluorescence imaging has quickly become an essential discovery tool for researchers, fast and scalable methods to classify and embed single-cell protein distributions in such images are lacking. Here, we present the design and analysis of the results from the competition Human Protein Atlas – Single-Cell Classification hosted on the Kaggle platform. This represents a crowd-sourced competition to develop machine learning models trained on limited annotations to label single-cell protein patterns in fluorescent images. The particular challenges of this competition include class imbalance, weak labels and multi-label classification, prompting competitors to apply a wide range of approaches in their solutions. The winning models serve as the first subcellular omics tools that can annotate single-cell locations, extract single-cell features and capture cellular dynamics.

The function of cellular systems is predominantly defined by the structure, amount, spatial location and interactions of individual proteins that collectively make up the proteome. While ongoing cell mapping efforts[1,2,3] are mainly based on single-cell sequencing technologies, protein localization is crucial to the understanding of biological networks. The expression and localization of proteins are well known to vary between healthy human cell types[4,3], even within genetically identical cell populations[5]. Therefore, protein subcellular localization needs to be attributed at the single-cell level. Tremendous technological progress in microscopy enables increasingly data-rich descriptions of cellular properties, including subcellular protein distribution. The increasing amounts of fluorescent image data necessitate better computational models that are capable of classifying the spatial protein distribution of single cells and ultimately enabling the investigation of how protein spatial regulation contributes to cellular function in health and disease.

The Human Protein Atlas (HPA) Subcellular Section has generated the first subcellular proteome map of human cells, consisting of a publicly available dataset of 83,762 confocal microscopy images detailing the subcellular localization of 13,041 proteins (HPAv21)[6] across a multitude of cell lines. Classifying the subcellular protein locations from these images is currently hampered by various technical challenges. First, around 55% of human proteins localize to multiple subcellular compartments, indicating that they might be involved in several biological processes[6–8], making it a multi-label classification problem. Second, the distribution of proteins in these compartments is highly disproportionate by up to five orders of magnitude (HPAv21)[6], causing an extreme class imbalance.

Third, training supervised models requires ground truth annotations of every single cell in a large set of images, which is time-consuming and costly. At the same time, the availability of image-level labels is much greater, which supports the need for weakly supervised models making use of image-level annotations.

Recent advances in artificial intelligence, especially deep neural networks (DNNs), have provided powerful algorithms to undertake these challenges. For example, class imbalance can be tackled at the data level, for example, by oversampling rare classes or under-sampling common classes[9], or at the classifier level, for example, by applying class weights[10,11] or using novelty detection[12]. Different variations of convolutional neural networks (CNNs) or fully connected neural networks are effective at solving segmentation and multi-class classification problems[13–17]. The key to success for deep learning models has been their ability to learn complex visual features from large quantities of labeled data (that is, supervised learning). Given that labeled data are expensive, there has been an increasing trend among deep learning researchers to develop models that can take advantage of unlabeled or weakly labeled data (that is, weakly supervised learning). For example, class activation map (CAM)-guided approaches[18] and weakly supervised adaptations of mask R-CNNs (region-based CNNs)[19,20] were designed to localize signal regions and assign precise object labels given coarse image-level labels.

In 2018 the Human Protein Atlas Image Classification competition focused on crowdsourcing machine learning solutions for multi-label image classification for a class-imbalanced dataset[15]. The winning solutions greatly improved the state of the art and have enabled novel insights into cellular architecture, such as the discovery

[1]Science for Life Laboratory, School of Engineering Sciences in Chemistry, Biotechnology and Health, KTH – Royal Institute of Technology, Stockholm, Sweden. [2]Department of Bioengineering, Stanford University, Stanford, CA, USA. [3]Changsha, China. [4]Copenhagen, Denmark. [5]Innopolis University, Innopolis, Russian Federation. [6]Research and Development Department, Guangdong Longsee Biomedical Corporation, Guangzhou, China. [7]LYSEWIRED, Toulouse, France. [8]Nvidia, Santa Clara, CA, USA. [9]Institute for Design Problems in Microelectronics of Russian Academy of Sciences, Moscow, Russian Federation. [10]Gideon Brothers, Zagreb, Croatia. [11]University Hospital 'Sveti Duh', Zagreb, Croatia. [12]Health Center Zagreb - West, Zagreb, Croatia. [13]Faculty of Electrical Engineering and Computing, University of Zagreb, Zagreb, Croatia. [14]Department of Pathology, Stanford University, Stanford, CA, USA. [15]Chan Zuckerberg Biohub, San Francisco, CA, USA. ✉e-mail: emmalu@stanford.edu

of a sub-pattern in the nucleolus[21] or construction of a multiscale hierarchical cell model by using the image features derived from the classifier[22]. It is a great demonstration of how competitions can be used as a tool to perform large-scale investigation of state-of-the-art methods by defining a clear research problem in the setting of a competition. As a continuation, deep learning models for generating single-cell protein labels will provide higher-quality feature maps, and enable studies of localization heterogeneity and dynamics in single cells and potentially facilitate the discovery of new subtle, rare or transient patterns that cannot reliably be observed across entire cell populations. This motivated us to host another competition to crowdsource solutions for using our HPA image dataset annotated with image-level labels in a weakly supervised setting to generate single-cell-level protein localization labels.

In this paper we present the design and analysis of the results from the competition Human Protein Atlas – Single-Cell Classification hosted on the Kaggle platform (https://www.kaggle.com/c/hpa-single-cell-image-classification). Given images, image-level labels and pretrained segmentation models, the participants were required to submit a solution able to generate a segmentation mask and single or multiple labels for each single cell in every image in a hidden dataset. Over 3 months, 757 teams registered a total of 19,058 submissions. The top-ranking teams were awarded with cash prizes: US$12,000, $8,000 and $5,000 for first, second and third place, respectively. The winning models were able to capture relevant biological features, and greatly improved the state of the art for classification of subcellular protein patterns. Here, we present the competition design, statistical analyses of the solutions and visualizations of the winning models to shed light on the considerations for designing multi-label single-cell pattern classification models based on weak and noisy training data. We also discuss potential applications of the winning solutions.

## Results

**Competition design and evaluation metrics.** The aim of this competition was to develop computational models to classify subcellular protein localization patterns for single cells in microscopy images, given a training set with only image-level labels (Fig. 1a). Successful submissions consisted of an individual segmentation mask and the protein localization labels (one or more of 19 classes) for every cell in each image of the test dataset. A training and test dataset was assembled to facilitate effective model training and to alleviate somewhat the extreme class imbalance in the HPA image dataset (Methods). The training dataset consisted of 21,806 confocal microscope images of 17 different human cell lines and proteins encoded by 7,807 different genes. Furthermore, participants had the possibility to use any publicly available external dataset or pretrained model, including the full HPA Subcellular Section dataset (HPAv20: 82,495 images)[6] (Supplementary Tables 1 and 2), as well as the pretrained in-house HPACellSegmentator (https://github.com/CellProfiling/HPA-Cell-Segmentation) or any other segmentation tool (for example, Cellpose[23]). Each image corresponds to a field of view consisting of multiple cells (average, 20 cells per image) where the protein of interest has been visualized together with three reference markers for the nucleus, microtubules and endoplasmic reticulum (ER). The images in the training dataset were annotated

with the standard HPA annotation pipeline[6,24], in which one or multiple labels were assigned to each image by assessing the localization patterns of all of the cells. The labels for each cell in the image are therefore not guaranteed to be precise because of single-cell heterogeneity. That is, in the same image of a genetically identical population, individual cells can have different protein localization patterns (Fig. 2). This makes the data weakly labeled and the classification task weakly supervised. To evaluate the performance of the participating models, a test dataset of 41,597 cells in 1,776 images with pronounced single-cell heterogeneity was manually annotated for single-cell labels (Methods) and used as the test dataset (Supplementary Tables 1 and 2). The test set was divided into a public test set with 31% of the images ($n = 559$) and a private test set with 69% of the images ($n = 1,217$), which has never been published before (Supplementary Table 1). Importantly, the participants can only see the images in the public test set and do not have access to any single-cell labels.

In contrast to our previous Kaggle competition[15], we chose to make this a code competition, in which participants were required to predict on test images by executing their source code via the Kaggle Notebook environment. This rule guarantees the reproducibility of the predictions and limits complicated solutions that would exceed inference resources. The submissions were evaluated using mean average precision[25] (mAP) at a mask-to-mask intersection over union (IOU) threshold of 0.6 across 19 classes (Methods). Importantly, during the competition, participants could see their rankings only on the public leaderboard, which was based on their mAP scores in the public test set. Only at the end of the competition was their performance on the private test set revealed on the private leaderboard, which determines their final rankings. The winning solutions were based on the scores on the private leaderboard.

**Participation and performance.** Over 105 days, 991 participants formed 757 teams and provided a total of 19,157 submissions. The top 10 teams made more submissions per day than the average of all of the teams (3.3 versus 0.2 submissions per day). A total of 56% of the participants in the top 10 teams (18 of 32) were grandmasters (the highest possible rank in Kaggle). Two of the top 10 teams did not have grandmasters but instead had domain knowledge experts (for example, people who had a PhD in biochemistry or were physicians), emphasizing the difficulty of this competition.

The top-ranking team reached an mAP of 0.59 on the public leaderboard and 0.57 on the private leaderboard. Analysis of class performance of the top 50 teams showed a wide spread, from an average precision of 0.0 to 0.8 (Supplementary Tables 3 and 4). In general, classes with distinct visual patterns or more training samples had higher average precision (AP) scores, such as Nucleoplasm (AP $0.68 \pm 0.10$), Nuclear Membrane (AP $0.68 \pm 0.10$), Microtubules (AP $0.63 \pm 0.09$) and Nucleoli (AP $0.63 \pm 0.09$). Classes with fewer training samples, highly varied location distribution and merged classes such as Negative (AP $0.34 \pm 0.11$), Centrosome (AP $0.33 \pm 0.07$), as well as Vesicles and cytosolic punctate pattern (AP $0.31 \pm 0.06$) achieved lower AP scores. Mitotic Spindle had the highest spread of performance in the top 50 teams (AP $0.38 \pm 0.23$), which probably stems from the approach of manual labeling of rare classes that some teams adopted (Supplementary Table 5). Interestingly,

**Fig. 1 | Challenge overview. a,** For training, participants were given access to a well-balanced training set and the public HPA dataset, which consists of fluorescent images with four channels and corresponding image-level labels. To evaluate the solutions' performance, a private (hidden) dataset of 1,776 images containing 41,000 single cells with corresponding single-cell labels was used. **b,** Overview of the main approaches: image-level models take in images and predict image-level labels, which can be combined with CAMs of different classes and segmented cell regions to give the final cell-level labels. Image-level models can also be trained on whole images, but predict on bag-of-cell tiles at inference to produce the cell labels. Cell-level models take in single-cell crops and predict single-cell labels. **c,** Number of images and cells per class in the training set and test set. **d,** Violin plot of the score distribution per label for the top 50 teams, ordered by decreasing cell count. $n = 50$ teams for each violin. The minimum, mean, percentile, and maximum values are noted in Supplementary Fig. 3. Ves. punctate, vesicles and punctate pattern. Scale bar, 10 μm.

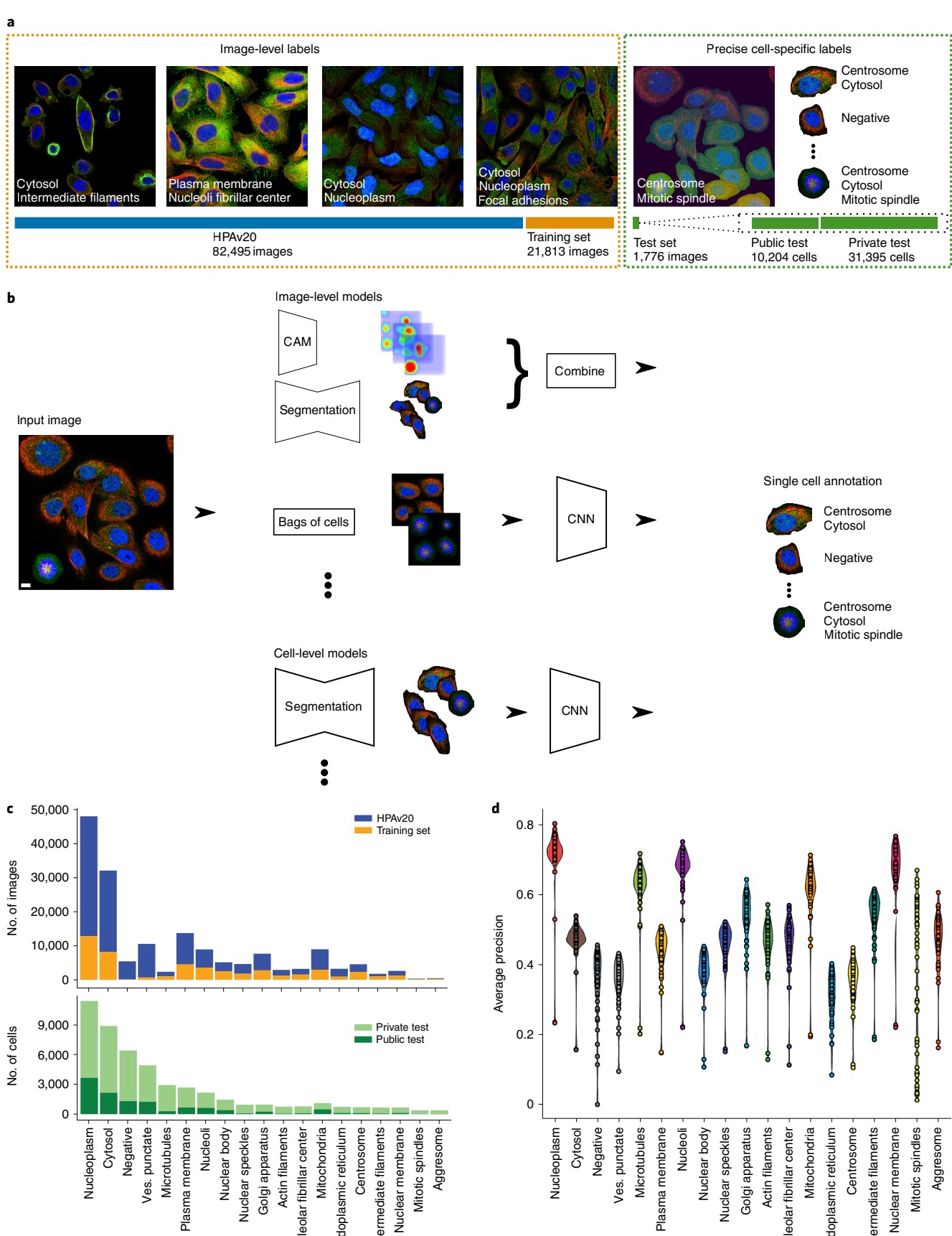

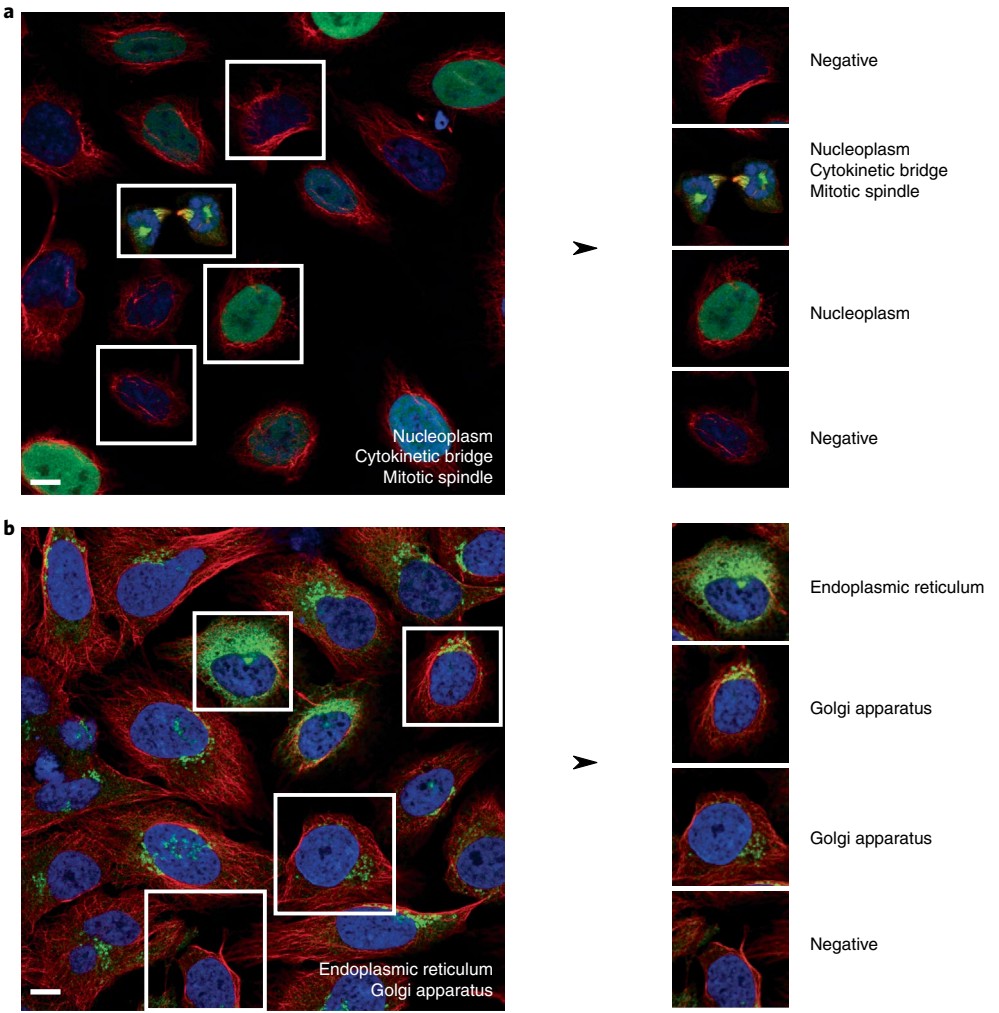

**Fig. 2 | Examples of single-cell spatial heterogeneity, in which each single cell does not inherit all of the image-level annotations. a**, The subcellular distribution of TPX2 microtubule nucleation factor in U-2 OS cells is labeled as Nucleoplasm, Cytokinetic bridge and Mitotic spindle. **b**, The subcellular distribution of Laminin subunit γ-2 is expressed in Endoplasmic reticulum and Golgi apparatus in U-2 OS cells. Scale bars, 10 μm.

ER, which was provided as a reference channel, achieved a medium performance in the public test set (AP $0.42 \pm 0.10$) and the worst performance in the private test set (AP $0.25 \pm 0.07$). This difference could be attributed to the use of 3- or 4- image channel inputs by the teams.

**Overview of approaches.** Most solutions used one of two main approaches, hereby referred to as 'cell-level models' and 'image-level models' (Fig. 1b and Supplementary Table 6). Image-level models take in whole images and predict image-level locations. To obtain single-cell labels, teams either combined activation maps of individual classes with segmented cell regions or used a bag-of-cells approach, which concatenates augmentations of one single cell into an image input. Cell-level models require segmented cell inputs to produce single-cell labels. The solution complexity varied greatly, with, for example, the ensembling of four single models by team 4, and 18 single models by team 2. Most teams used traditional networks as backbones and did specific modifications to achieve higher performance on this specific dataset. Popular backbone architectures include the InceptionNet[26] family, multiple EfficientNets[27], DenseNet[28], ECA-Net[29] and different variations of vision transformers such as Swin transformer[30] and DeiT[31]. The most popular approach for image-level models was Puzzle-CAM[32] and its

modifications. The activation of a CNN on feature maps of an image focuses on the most discriminative instance of a class despite the existence of many instances, in this case cells. Due to the design of this competition, with highly heterogeneous images in the test set, high scores required correct identification of all single cells, not only the most discriminative cells. Therefore, team 1 came up with a new approach for fair activation based on Puzzle-CAM, termed the Fair Cell Activation Network, whereby the network was forced to pay attention to single cells because the training patches contain a mix of original crops from full images or masked out single cells (Supplementary Figs. 4,5 and Supplementary Note 1). The results of image-level prediction and cell-level predictions were combined either by addition or multiplication.

To handle the extreme class imbalance, several teams used weighed loss or oversampling. The most popular loss in this competition was focal loss[33], which addresses the large class imbalance in the training and validation datasets by decaying the scaling factor when the confidence of that class increases. The addition of external data and hand-labeled data also improved the performance for rare classes such as Mitotic Spindle. Given that most teams made use of the HPACellSegmentator (Supplementary Table 5), with several community-developed improvements such as faster run-time, no significant correlation can be seen between segmentation

**Table 1 | Summary of the top solutions**

| Rank | Team name | mAP score | No. of submissions | Overview of methods |
|------|-----------|-----------|--------------------|---------------------|
| 1 | bestfitting | 0.5667 | 480 | Fair cell activation network, rule-based cell labeling based on image-level predictions, transformer |
| 2 | [red.ai] | 0.55328 | 459 | Cell-level models, image-level models, custom loss |
| 3 | MPWARE & ZFTurbo & Dieter | 0.54995 | 500 | Puzzle-CAM, image-level models, out-of-fold predictions |
| 4 | MILIMED | 0.54389 | 258 | Data-centric approach, manual labels, cell-level models |
| 5 | narsil & David & tito | 0.54243 | 518 | Image-level model predicts images made up of augmentations of one cell |
| 6 | scumed&Mitotic Spindle | 0.54108 | 435 | Image-level models, cell-level models |
| 7 | PFCell | 0.54053 | 162 | Image-level models, CAM for cells, pseudo-labels |
| 8 | Guanshuo Xu | 0.53898 | 62 | Image-level label multiplied with CAMs |
| 9 | AllDataAreExt & Galixir | 0.53557 | 488 | Cell-level models, image-level models |
| 10 | [RAPIDS.AI] Cell Game [Rist] | 0.53503 | 166 | Cell-level models, manual label for rare class, treat weak label as noise |
| 11 | Silvers | 0.53287 | 411 | Cell-level models, image-level models, pseudo-labels |
| 12 | Andrew Tratz | 0.53049 | 134 | Image-level models, cell-level models |
| 13 | CVSSP + forecom.ai | 0.52717 | 202 | Manual label for rare class, image-level models, cell-level models |
| 15 | Fumihiro Kaneko | 0.5186 | 144 | Image-level models, pesudo-labels from CAM |
| 16 | Looking for the lost cell | 0.51711 | 431 | Pseudo-labels, image-level models, cell-level models |
| 18 | yuvaramsingh | 0.51559 | 162 | Image-level labels, transformer |
| 20 | Da Yu | 0.50626 | 88 | Cell-level models, image-level labels |
| 21 | Alexander Riedel | 0.50249 | 238 | Image-level models multiplied by CAMs |
| 22 | cool-rabbit | 0.50106 | 399 | Pseudo-labels, cell-level models |
| 23 | Raman | 0.50016 | 150 | Pseudo-labels, gradient accumulation |
| 24 | Shai | 0.49222 | 209 | Image-level model multiplied by CAMs |
| 27 | Mikhail Gurevich | 0.48319 | 85 | Cell-level models (Puzzle-CAM), image-level models, pseudo-labels |
| 28 | Histopathological Challenger | 0.48066 | 26 | Cell-level models, image-level models, pseudo-labels |
| 32 | Quoc-Hung To | 0.46893 | 152 | Cell-level models, image-level models |

score and average precision score (Methods, Supplementary Fig. 1b). However, some teams applied post-processing techniques to detect border cells, which had a positive impact on performance: some teams created new DNNs (Supplementary Notes 1 and 3) or used special heuristics (Supplementary Table 5 and Supplementary Note 2).

All of the solutions in this competition are based on DNNs. The top four teams had approaches that covered the scope of this competition, therefore they were chosen for further in-depth analysis.

**Strategies of the winning teams.** To evaluate the impact of individual model components, each of the top four teams performed an ablation study (Supplementary Tables 6–9 and Supplementary Notes 1–4). All four teams found that carefully ensembling different pipelines of smaller models, trained with different loss functions and augmentations, resulted in higher scoring solutions. Not all ensembling worked; for example, team 4 found that checkpoint ensembling had a negative impact on the score (Team 4, Supplementary Table 9). Due to the restrictions imposed by code competitions, the teams made strategic choices to their final models such as using less augmentations in order to save inference time. For future deployment of these models it is possible to compress a large ensemble of models into a single-cell-level model trained on out-of-fold prediction without sacrificing performance greatly (experiment 23 and 24, team 3, Supplementary Table 9). Teams 1 and 4 also noted that image resolution had little impact on the final score, while cell weighing, border and outlier detection had a positive impact on the final score (Supplementary Tables 6 and 8). A generalized summary of the winning strategies is given in Table 1.

**Class activation mapping.** To assess whether the top models pay attention to biologically relevant subcellular locations when assigning probability for individual classes, gradient-weighted CAM[34] was used to visualize the coarse heatmap of regions important for the prediction. Figure 3 shows the comparison between CAMs for a few representative single and multi-labeled cells, including CAMs from cell-level models (Fig. 3a,b, from inceptionv3 of team 1) and image-level models (Fig. 3c, from a Siamese network[35] with a SEResNext50 backbone of team 3). Overall, the top models produced visual attention patterns that focused on regions corresponding to the staining of specific subcellular structures. Namely, both the image-level and the cell-level models reflect biologically meaningful features. For single-labeled cells, CAMs did not have a complete overlap with the protein channel signal (IOU = 0.28 with $n = 13,577$ cells, Supplementary Fig. 10 and Supplementary Table 10), but approximately delineated the most discriminative region for that class pattern (Fig. 3a). A similar trend was observed for multi-class patterns, and CAMs seem to be complementary to each other (Fig. 3b). By contrast to the cell-level model, multi-class CAMs from image-level models did overlap with each other, approximately delineating less precise regions of attention across the whole cells (Fig. 3c).

**Single-cell variability in the HPA Subcellular Atlas.** Uniform Manifold Approximation and Projection (UMAP)[36] was used to visualize the single-cell features produced by the winning cell-level model. By visualizing feature clusters, we can see how well the network features represent and separate cellular structures and compartments. First, single-cell features and class probability were extracted for the 12,770 human proteins mapped in the Cell

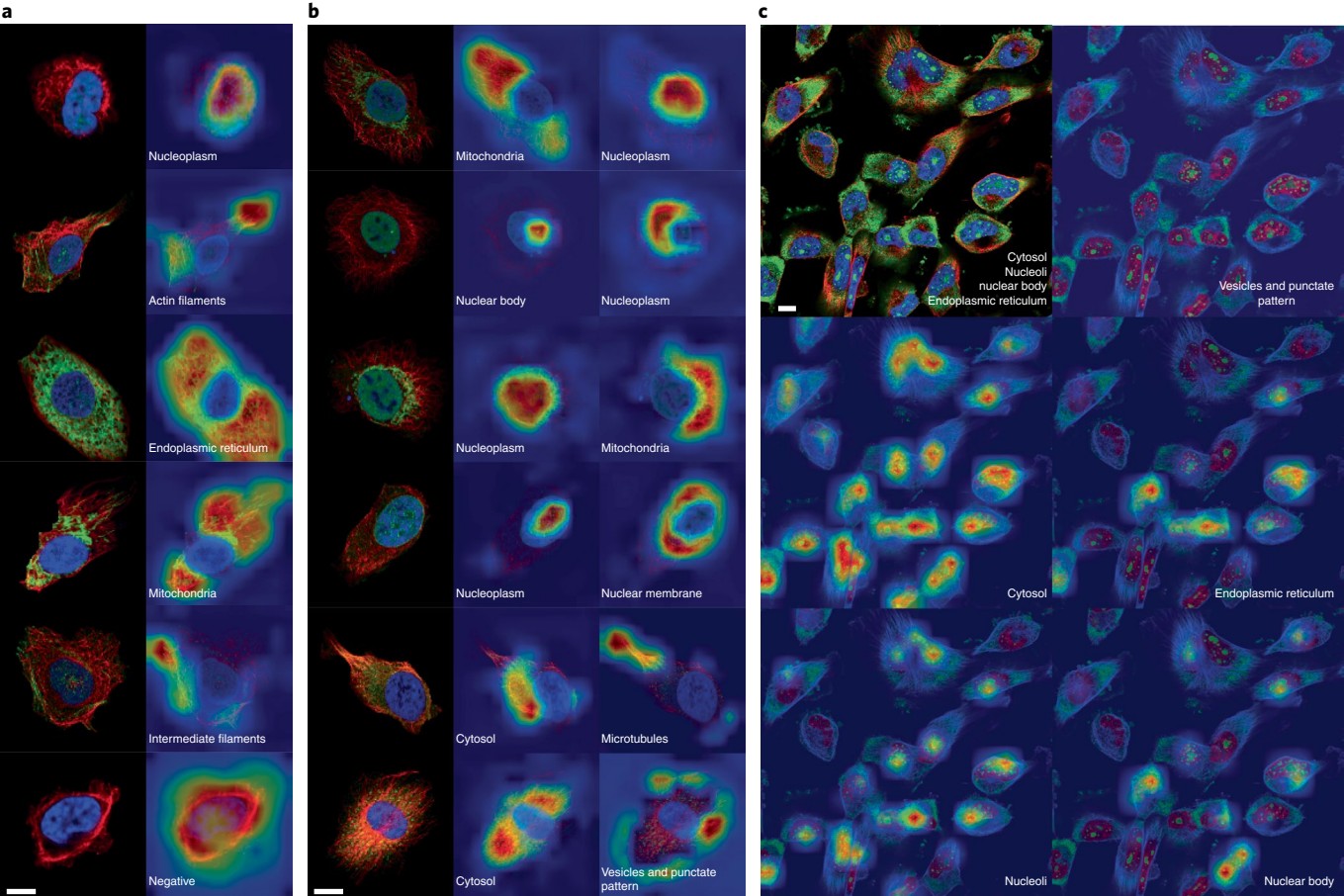

**Fig. 3 | Spatial attention of models. a**, For cells with a single label, the CAMs from inceptionv3 from team 1 highlight distinct features of the respective class. **b**, For cells with multiple labels (that is, protein in two or more cellular compartments), the CAMs of individual classes highlight relevant cellular regions. The combined CAMs approximately covered the area of the green signal. **c**, The image-level model captures more general CAMs for each of the image-level classes. For the same image of RPL4 expression in the cell line U-251 MG, CAMs for Cytosol, Endoplasmic reticulum, Nucleoli and Nuclear bodies highlight the approximate regions of each cell that has that pattern. These activation maps are not inclusive in that not all of the cells with the correct label are highlighted. However, the activation map of Vesicles and punctate pattern, which is in part overlapping the activation map of Endoplasmic reticulum, is empty, indicating that image-level models can capture biological relevant features and distinguish between overlapping patterns. Scale bars, 10 μm.

Atlas of the Human Protein Atlas (HPAv20). The feature embedding was reduced by UMAP and colored by image-level labels with high prediction probability (Methods). Clusters with distinct subcellular patterns and high scores such as Microtubules, Nuclear Membranes and Nucleoli separate nicely (Fig. 4a). Patterns in the same meta-compartment are in closer proximity to each other, with nuclear and cytoplasmic meta-clusters clearly separated. This indicates that the model also takes the cellular location of the pattern matters into account for classification. An example of similar punctate patterns that end up in different meta-clusters are nuclear bodies (nuclear meta-cluster) and vesicles (cytoplasmic meta-cluster). Examples of different patterns that still form such close-proximity clusters include Nuclear bodies, Nucleoli and Nucleolar fibrillar center, which are all compartments of the nuclear meta-cluster, or Actin, Endoplasmic reticulum, Golgi apparatus and Mitochondria, which are all compartments of the cytoplasmic meta-cluster. (Fig. 4a). Most cells with multiple labels (that is, when the protein localizes to multiple cellular compartments) naturally lie between clusters of cells with single class labels (Fig. 4b). Vesicles and Punctate pattern include a variety of trafficking and signaling organelles such as peroxisomes, endosomes, lysosomes, lipid droplets and cytoplasmic bodies. Features learned for this class are more likely to be found within the boundaries of classes representing major

compartments such as Cytosol, Golgi apparatus and Endoplasmic reticulum, possibly reflecting the highly dynamic properties of these trafficking vesicles (Fig. 4c).

Differences in protein expression patterns in genetically identical cells can be attributed to several factors, such as the cell cycle, metabolic states or different signaling functions. By comparing the single-cell labels derived from the winning model with the HPAv20 image labels, we can assess the extent of spatial heterogeneity of protein expression in the HPA. Out of all of the multi-localizing proteins (that is, proteins with multiple location labels), 452 are present only in a single location per single cell, indicating precisely coordinated temporal translocation events. These proteins are heavily enriched for different cell cycle processes especially related to mitosis and include several known cell cycle proteins[37] (Supplementary Fig. 10). Translocation of proteins involved in cell division is highly expected because the process involves the formation of mitotic structures such as the mitotic spindle and the cytokinetic bridge, which are seen only in certain stages of the cell cycle. Proteins involved in transcriptional regulation are also known to translocate, such as, for example, transcription factors, which may translocate in and out of the nucleus in response to extracellular or intracellular signals (Supplementary Fig. 10b,c). Similarly, the enrichment of proteins involved in endocytosis and cell adhesion (Supplementary

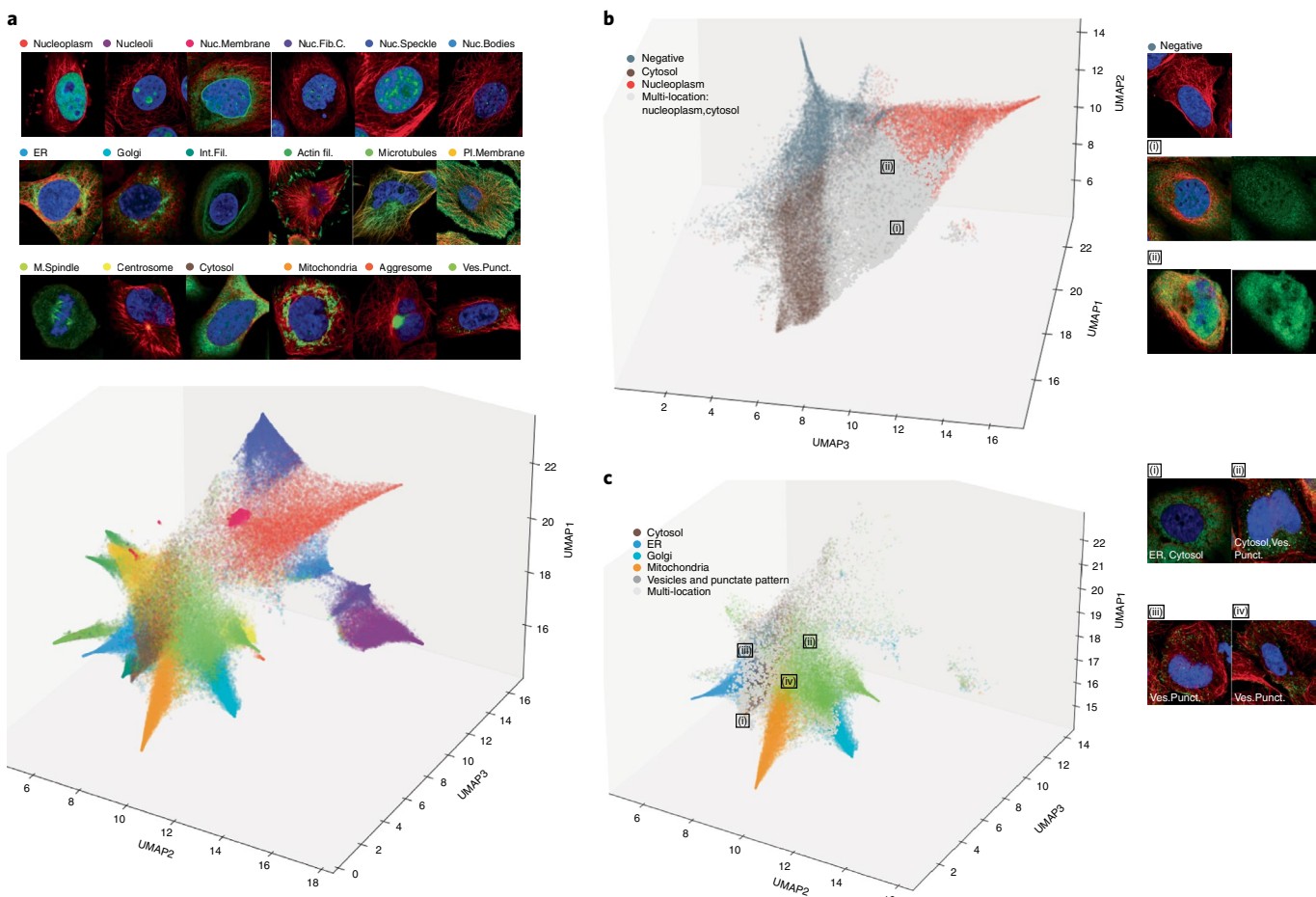

**Fig. 4 | Visualization of learned single-cell features. a**, Overview of all single-cell clusters (except Negative). Notably, the patterns organize into cytosolic (bottom-left) and nuclear (top-right) meta-clusters. **b**, Cells with both the Cytosol and the Nucleoplasm labels lie between the two respective clusters. The majority of negative cells cluster into a separate corner with a slight overlap with Cytosol and Nucleoplasm clusters. **c**, Cells labeled with Vesicles and punctate pattern (Ves. Punct.) in the cytosol, endoplasmic reticulum (ER) and the Golgi apparatus cluster together at the opposite end to the nuclear patterns in the three-dimensional UMAP plot. Cells labeled Vesicles can be found in these other clusters, capturing the cellular dynamics of the class. Actin fil., Actin filaments; Golgi, Golgi apparatus; Int. Fil, Intermediate filaments; M. Spindle, Mitotic spindle; Nuc. Bodies, Nuclear bodies; Nuc. Fib. C., Nucleolar fibrillar center; Nuc. Membrane, Nuclear membrane; Nuc. Speckle, Nuclear speckle; Pl. Membrane, Plasma membrane.

Fig. 10d) may reflect the movement of membrane proteins along the secretory pathway (that is, plasma membranes, the Golgi apparatus, endoplasmic reticulum and vesicles).

## Discussion

Organizing a citizen science competition with the aim to produce useful models requires deep knowledge of the data, the current state of the art of relevant DNNs, and a clear view of the desired aim of the competition (that is, the area where innovation is needed). With this in mind, the challenge description, training and test datasets and evaluation metrics must be assembled to best assist participants to reach the competition goal under the given restrictions. There is a risk that the winning solutions are ensembles of weak learners, much too complex to be deployed, or that the models are unable to capture relevant biological features, making them irrelevant for future use. To mitigate these pitfalls we need to guarantee fair metrics and fair data. For metrics, we chose mAP, which focuses on overall model performance, rather than the threshold-specific performance for each class in the case of the more commonly used F1 score. For data, we ensure that the single-cell labels are accurate (90% agreement between human annotators, Methods) and that the dataset reflects the normal physiological proportion of subcellular patterns. Given that the competition is time constrained, we made

the conscious choice to reduce the magnitude of class imbalance from five to two orders of magnitude (Supplementary Table 1). And given that no single-cell labels were provided for training, participants relied on their public leaderboard score (that is, the public test set) for validation. The careful division between the public and private test sets ensures the same proportional class imbalance. This led to a very minor shake-up in the final ranking, supporting our fair competition design of datasets and metrics.

Throughout the competition we observed great community spirit and an open science approach. Participants shared observations, technical data processing tips and biological knowledge with one another in public notebooks and discussions. For example, once several members had found boosted performance when combining cell-level models and image-level models, many teams followed suit. Similarly, the enhanced version of the HPACellSegmentator was quickly adopted.

The existence of rare protein patterns causing extreme class imbalance, protein multi-localization and weak labels presented the main challenges for this competition. Although the first two challenges are common for vision models, this competition is the first example of multi-label image classification based on weak and noisy biological data. The key to high performance in this competition was to find approaches that spread attention evenly to every cell, not

only the most discriminative one in the image. Participants tackled this in different ways, such as minimizing differences between features from sub-regions to the whole image with the Puzzle-CAM[32] approach, using attention-based networks such as transformers[30,31] or adding attention blocks[38] after obtaining feature maps in the traditional CNN architectures. Because the HPACellSegmentator was provided, some participants turned this competition into object detection while others considered it as weakly supervised semantic segmentation.

As with any other machine learning work, the winning models carry the bias of the training data, and the performance in some classes is better than in others. Although the best single model performance reaches an average precision of 0.6 for half of the classes, even 0.8 for some classes (Supplementary Table 3), rarer classes differed considerably from human expert performance (accuracy ~0.9 based on agreement between multi-annotators). Developments in self-supervised[39], unsupervised[40] and few-shot[41] learning could potentially tackle high class imbalance and rare class or novelty detection to a much greater extent. However, despite these limitations, the winning models still demonstrated an ability to focus attention on the subcellular regions in which the proteins lie, and we see great advances in cell-level model attention compared with image-level model attention (Fig. 3b,c). Most importantly, the model can be used to refine single-cell labels up to the scale of millions, way beyond the scope of human expert labor.

Precise single-cell classification and good local attention of multi-label single-cell patterns enable the exploration of the dynamics of protein localization in single cells. Previously, the output of the image classification challenge enabled the modeling of a unified hierarchical map of eukaryotic cell architecture[42]. One can imagine that a model able to capture the features of single cells with much finer resolution will surely be powerful in mapping the cellular structure and facilitating new discoveries in cell biology through modeling the multi-scale proteome architecture. It is expected that the learned feature embedding produced by the winning models will be able to shed light on single-cell spatial variability within images, protein spatial expression heterogeneity across cells and biological processes like the cell cycle or cell migration, and provide a better understanding of dynamic protein functions in different organelles.

## Online content

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

## Methods

**Segmentation model.** For segmentation of single cells, a pretrained DPN (dual path network) U-Net, which was the winning architecture in the 2018 Data Science Bowl[43], was further trained on HPA data. For each image, Nuclei, Microtubules and Endoplasmic Reticulum channels were used as 3-channel input and manually annotated cell masks were used as ground truth. Each channel had a frame size of 2,048 × 2,048 pixels and was acquired with a ×63 objective with a pixel resolution of 80 nm.

The DPN U-Net neural network was trained and validated on a dataset of 266 HPA Cell Atlas images from 28 cell lines, which were split into training (257 images) and validation (9 images) datasets. During inference, a pretrained DPN U-Net for nuclei segmentation[43] was used to generate nuclei mask images, followed by customized post-processing steps with the watershed algorithm to obtain the individual nuclei. Then, using the DPN U-Net trained on our cell images, we obtained the cell masks. The nuclei obtained in the first step were combined with the cell mask in a watershed-based post-processing step to produce the segmented cells. The activation function was softmax, and the loss function was a combination of BCE (binary cross entropy) and Dice Loss. The model was trained using one Nvidia GeForce RTX 2070 graphics card until convergence, which took approximately 1.5 h (with pretrained weights). The source code for the DPN U-Net segmentation model is available at https://github.com/cellProfiling/hpa-cell-segmentation. Data and pretrained weights are available at https://zenodo.org/record/4665863.

**Single-cell annotation pipeline.** The annotation of ground truth segmentation masks and single-cell labels followed a two-step procedure. In the first step, a pretrained DPN U-Net[43]-based model (HPACellSegmentator; https://github.com/CellProfiling/HPA-Cell-Segmentation), with weights trained on HPA public images, was used to generate baseline segmentation masks for all images in the test set. Every image was subsequently manually inspected and adjusted for inaccurate cell masks. In the second step, individual cells were annotated by experts from the HPA team. The cells were presented to the annotator one by one, and no metadata were shown except the image-level labels. Each cell was shown in the bounding box region of its segmentation mask, padded by 50 pixels on all sides. The order of cells to annotate was randomized so that cells in the same image were not presented in sequence. A total of 10% of all cells were annotated by multiple people to control for annotation consistency. In this overlapped single-cell set, labels are 90% consistent between multiple annotators. (This agreement is much higher compared with chance overlap, for which the probability of two annotators choosing the same organelle label out of 30 labels is 0.1% and that for choosing two of the same organelle labels out of 30 is 0.0001%, while some cells have up to four annotators and some cells have more than two labels, or 67% agreement between four pathologists for surgical pathology reports[44].)

**Label processing and final competition dataset.** In the annotation process, each single cell was classified into one or more of the 30 organelle labels or Negative/Unspecific. These labels were: Nucleoplasm, Nuclear membrane, Nucleoli, Nucleolar fibrillar center, Nuclear speckles, Nuclear bodies, Endoplasmic reticulum, Golgi apparatus, Peroxisomes, Endosomes, Lysosomes, Intermediate filaments, Actin filaments, Focal adhesion sites, Microtubules, Microtubule ends, Mitotic spindle, Centrosome, Lipid droplets, Plasma membrane, Cell Junctions, Mitochondria, Aggresome, Cytosol, Cytoplasmic bodies, Rods and Rings, Cleavage furrow, Centriolar satellite, Vesicles, Mitotic chromosome, and Negative–Unspecific. All images with more than 50% of the cells classified as Negative–Unspecific or more than 50% of defective cells were removed. All images in which all single-cell labels correspond exactly to image-level labels were also removed.

To simplify the challenge and balance the class distribution we prioritized and grouped classes that are functionally and spatially similar into 19 classes: Nucleoplasm, Nuclear membrane, Nucleoli, Nucleolar fibrillar center, Nuclear speckles, Nuclear bodies, Endoplasmic reticulum, Golgi apparatus, Intermediate filaments, Actin filaments (consisting of Actin filaments and Focal adhesion sites), Microtubules, Mitotic spindle, Centrosome (consisting of Centrosomes and Centriolar satellites), Plasma membrane (consisting of Plasma membrane and Cell junctions), Mitochondria, Aggresome, Cytosol, Vesicles and punctate cytosolic pattern (consisting of Vesicles, Peroxisomes, Endosomes, Lysosomes, Lipid droplets and Cytoplasmic bodies) and Negative.

The final test set consisted of 1,776 images of 41,597 single cells from 17 human cell lines of varying morphology. The cell lines were A-431, A549, EFO-21, HAP1, HEK 293, HUVEC TERT2, HaCaT, HeLa, PC-3, RH-30, RPTEC TERT1, SH-SY5Y, SK-MEL-30, SiHa, U-2 OS, U-251 MG and hTCEpi.

The training set was constructed by sampling images from the 17 listed cell lines from the training set of the previous Kaggle challenge[15]. The image-level labels were grouped and reindexed similarly to the single-cell labels in the test set. The final training set consists of 21,813 images.

**Evaluation metrics.** The metrics used to rank the performance of the teams in the Kaggle competition was mAP[25] at a mask-to-mask IOU threshold of 0.6 across 19 classes in the challenge. In short, for each image, all predicted cell masks are matched to all ground truth cell masks. If two masks have IOU > 0.6 then this

pair is considered matched. The true and false positives and negatives are counted for all matched pairs. All labels for non-matched detections are considered false positives.

For each class, precision and recall are calculated:

$$\text{Precision} = \frac{\text{True positives}}{\text{True positives} + \text{False positives}}$$

$$\text{Recall} = \frac{\text{True positives}}{\text{True positives} + \text{False negatives}}$$

Precision is represented as a function of recall $r$, and average precision computes the average value for precision $p(r)$ over the entire interval of recall from $r = 0$ to $r = 1$: (ref. [24]). Details about implementation can be found at https://storage.googleapis.com/openimages/web/evaluation.html#instance_segmentation_eval.

$$\text{Average precision} = \int_0^1 p(r)\, dr \text{ (Definition 1 from ref. 22)}$$

The final score is the mean of the average precision scores of all 19 classes.

**Intersection over union.** The metric to measure the segmentation performance of the models is the IOU between the predicted segmentation and the ground truth segmentations.

$$\text{IOU}(p, g) = \frac{p \cap g}{p \cup g}$$

where $p$ is the predicted segmentation vector and $g$ is the ground truth vector. A predicted segmentation is considered correct if the IOU of the segmented cell to one of the ground truth segmentations is at least 0.6.

**Statistical analysis.** Plotting and statistical analysis were performed with Python 3, NumPy, Pandas, scikit-learn, seaborn, Matplotlib and plotnine. Overall participation and performance (members per team, submissions, public and private scores for each submission and so on) were extracted from Meta Kaggle (https://www.kaggle.com/datasets/kaggle/meta-kaggle). To test the correlation between mAP scores and segmentation IOU, Pearson correlation was performed. First, the IOU of matched cells (at a threshold of 0.6) was calculated for the best submission of the top 50 teams. (Most top teams chose two submissions for consideration for final scoring. For this analysis, the highest scored submissions were manually selected for each team that did not choose specific submissions.) For the same submissions, average precision per class and average precision per private and/or public test set were calculated using the TensorFlow Object Detection application programming interface (https://github.com/tensorflow/models/tree/master/research/object_detection, the same metrics as in the Kaggle leaderboard). The Pearson correlation between the mean IOU score and the mAP for all top fifty teams is $r = -0.3$ with a two-tailed $P = 0.02$. However, at d.f. = 48 the critical value of the Pearson correlation is 0.33, therefore we concluded that $r = 0.3$ is likely to occur by chance.

**Solution and approach summary.** The summaries of solutions and approaches were dependent on responses from participants in the post-competition survey, the Kaggle discussion forum and Kaggle kernels. Table 1 is largely generalized, and more details are given in Supplementary Note 5 and Supplementary Table 5.

**Class activation map.** For each CNN model, Grad-CAM[34] was used to analyze which region of images each model is focusing on. For each model, we generated the CAMs from the appropriate convolutional block and overlaid the generated heatmap onto the original input image for illustration. As an example, CAMs were generated from team 1's inceptionv3 model.

**Single-cell prediction and UMAP.** A total of 83,763 images from the HPA Subcellular Section (HPAv20)[6] were preprocessed as input to the trained network, resizing the images (1,728 × 1,728 pixels, 2,048 × 2,048 pixels, 3,012 × 3,012 pixels) to 512 × 512 pixels. HPACellSegmentator was then used to produce single-cell masks in every image, yielding approximately 1.5 million cells. The cropped single cells were resized to 128 × 128 pixel patches and fed into the top performing inceptionv3 network from team 1 (bestfitting). The model predicted probabilities of each single cell belonging to each of the 19 classes. Prediction of 1.5 million cells took around 7 h on two TU106 [GeForce RTX 2070 Rev.A] GPUs (graphics processing units).

The single-cell features before the last densely connected layer were extracted so each single cell was condensed into a numeric vector of size 2,048. All together, a matrix of 1,500,000 × 2,048 was reduced to 1,500,000 × 3 by applying UMAP from umap-learn[36] (v0.5.2) with n_neighbor = 15, distance = 0.01, component = 3, metric = euclidean and n_epoch = 100000.

Single-cell annotations were determined by combining the single-cell prediction from the same network (inceptionv3) with the image annotation from the HPA Cell Atlas as follows: step 1, all single cells inherit all image-level labels

(il_labels); step 2, single-cell predictions are generated for each class (sc_labels); step 3, multiply the two labels il_labels × sc_labels and round up the product (given that il_labels is a sparse matrix, this multiplication will not add any new labels but instead remove low-certainty labels predicted by the models); and step 4, assign all cells with no labels as 'negative'.

A maximum of 10,000 cells were randomly sampled from each class to be visualized in Fig. 4 (160,000 cells in total). All cells in each class can be fully visualized using a web application deployed with the Imjoy platform[45], where users can interact with each cell and its corresponding image and metadata (https://imjoy.io/lite?plugin=https://raw.githubusercontent.com/CellProfiling/cellpro-imjoy-plugins/master/src/HPA-UMAP-single-cell.imjoy.html).

**Single-cell heterogeneity and gene set enrichment analysis.** To identify subtle single-cell heterogeneity within genetically identical populations (same cell line), the single-cell labels were compared with each other and with image-level labels. If all cells in an image carry the same set of labels, it is not heterogeneous. If the labels of the cells are different, it is counted as heterogeneity in that cell line. Cells for which the nuclei lie in the border of the image were not taken into account.

For proteins with multiple locations on image labels but which had only a single location at the single-cell level as predicted by the inceptionv3 model in the previous section, gene set enrichment analysis (GSEA) was performed. GSEA was done with gseapy v0.10.8 (wrapper for Enrichr[46–48]) against GO_Molecular_Function, GO_Biological_Process_2021, KEGG_2021, Reactome_2021. The background used was the human genome ('hsapiens_gene_ensembl'), with the cut-off at an adjusted $P$ value of 0.05 (Supplementary Fig. 10).

**Model ablation study.** Each of the top four teams conducted a series of ablation studies separately. For the top single model, depending on its specific architecture, some ablations will be performed. These can include input resolutions, changes in augmentation methods, changes in loss function, and changes in the model's backbone. mAP scores were calculated for the predictions from the perturbed model, then compared with those scores for the predictions from the unperturbed model.

**Reporting summary.** Further information on research design is available in the Nature Research Reporting Summary linked to this article.

## Data availability
The dataset used for the HPA competition is available at https://www.kaggle.com/c/hpa-single-cell-image-classification. The external dataset HPAv20 is publicly available at https://v20.proteinatlas.org/. A script for downloading the dataset is available at https://www.kaggle.com/lnhtrang/hpa-public-data-download-and-hpacellseg.

## Code availability
Source code used to produce the figures has been released under license at https://github.com/trangle1302/HPA_SingleCellClassification_Analysis. The teams' solutions are available at https://github.com/topics/hpa-challenge-2021.

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

## Acknowledgements
The authors thank all of the participants of the Human Protein Atlas (HPA) Image Classification competition and acknowledge the staff at Kaggle, especially P. Culliton and M. Demkin, for providing the competition platform that enabled this study. The staff of the HPA program provided valuable contributions, such as data storage and management, and J. Fall helped with project administrative tasks. The competition prizes were funded by Kaggle. Research funding was provided by the Knut and Alice Wallenberg Foundation (grant no. 2021.0346, 2021.0189, 2018.0172), the Erling Persson Foundation and the Swedish Research Council (grant no. 2017–05327), and Stanford startup funding to E.L.

## Author contributions
E.L. conceived the project. T.L., C.F.W., W.O. and E.L. designed the challenge framework. T.H.N.L. wrote the manuscript with inputs from all co-authors. E.L. and W.O. reviewed and edited the manuscript. T.H.N.L., W.O. and H.X. designed the software for annotation, curated the data and provided the segmentation tool. U.A., D.M., E.L., and J.M.K. annotated the single-cell labels in the test set. S.D., I.S.M., V.O., Y.X., E.B., C.H., R.A.S., N.B., V.B., A.B. and A.M. developed model code and descriptions for their teams. T.H.N.L. and C.F.W. performed the formal analysis and validation. E.L. acquired the funding and supervised the project. All authors read and approved the final manuscript.

## Funding

## Competing interests
The authors declare no competing interests.

## Additional information

**Correspondence and requests for materials** should be addressed to Emma Lundberg.

# nature research

# Reporting Summary

Nature Research wishes to improve the reproducibility of the work that we publish. This form provides structure for consistency and transparency in reporting. For further information on Nature Research policies, see our Editorial Policies and the Editorial Policy Checklist.

## Statistics

For all statistical analyses, confirm that the following items are present in the figure legend, table legend, main text, or Methods section.

| n/a | Confirmed | |
|---|---|---|
| ☐ | ☒ | The exact sample size (*n*) for each experimental group/condition, given as a discrete number and unit of measurement |
| ☐ | ☒ | A statement on whether measurements were taken from distinct samples or whether the same sample was measured repeatedly |
| ☐ | ☒ | The statistical test(s) used AND whether they are one- or two-sided<br>*Only common tests should be described solely by name; describe more complex techniques in the Methods section.* |
| ☒ | ☐ | A description of all covariates tested |
| ☐ | ☒ | A description of any assumptions or corrections, such as tests of normality and adjustment for multiple comparisons |
| ☐ | ☒ | A full description of the statistical parameters including central tendency (e.g. means) or other basic estimates (e.g. regression coefficient) AND variation (e.g. standard deviation) or associated estimates of uncertainty (e.g. confidence intervals) |
| ☐ | ☒ | For null hypothesis testing, the test statistic (e.g. *F*, *t*, *r*) with confidence intervals, effect sizes, degrees of freedom and *P* value noted<br>*Give P values as exact values whenever suitable.* |
| ☒ | ☐ | For Bayesian analysis, information on the choice of priors and Markov chain Monte Carlo settings |
| ☒ | ☐ | For hierarchical and complex designs, identification of the appropriate level for tests and full reporting of outcomes |
| ☐ | ☒ | Estimates of effect sizes (e.g. Cohen's *d*, Pearson's *r*), indicating how they were calculated |

*Our web collection on statistics for biologists contains articles on many of the points above.*

## Software and code

Policy information about availability of computer code

| Data collection | We used Leica SP5 confocal microscope (DM6000CS) equipped with a 63× HCX PL APO 1.40 oil CS objective (Leica Microsystems, Mannheim, Germany) and Leica Application Suite Advanced Fluorescence (LAS AF, 2.6.0 build 7266) to collect images used in this work. However the image collection (including experimental procedures and samples/reagents such as cell line authentication and antibody validation) is described in previous publications and is not part of this work. |
|---|---|
| Data analysis | For the competition dataset we mainly used Python 3 (with libraries such as numpy 0.17.0, pandas 0.24.2, scikit-image 0.15.0, scikit-learn 0.21.3, scipy 1.1.0, matplotlib 2.2.2, seaborn 0.9.0). Pretrained segmentation model is available at https://github.com/cellProfiling/hpa-cell-segmentation. Software libraries used by the top teams are described in their own source code available at https://github.com/topics/hpa-challenge-2021 |

For manuscripts utilizing custom algorithms or software that are central to the research but not yet described in published literature, software must be made available to editors and reviewers. We strongly encourage code deposition in a community repository (e.g. GitHub). See the Nature Research guidelines for submitting code & software for further information.

## Data

Policy information about availability of data

All manuscripts must include a data availability statement. This statement should provide the following information, where applicable:

- Accession codes, unique identifiers, or web links for publicly available datasets
- A list of figures that have associated raw data
- A description of any restrictions on data availability

The dataset used for the HPA competition is available at: https://www.kaggle.com/c/hpa-single-cell-image-classification

# Field-specific reporting

Please select the one below that is the best fit for your research. If you are not sure, read the appropriate sections before making your selection.

☒ Life sciences  ☐ Behavioural & social sciences  ☐ Ecological, evolutionary & environmental sciences

For a reference copy of the document with all sections, see nature.com/documents/nr-reporting-summary-flat.pdf

# Life sciences study design

All studies must disclose on these points even when the disclosure is negative.

| | |
|---|---|
| Sample size | Sample size of training and testing dataset, as well as public HPA was detailed in Figure 1A and Methods. For analysis, top 50 teams were chosen as representation of overall participating team's performance. Based on the available solution write-ups, we chose the top 4 teams as representation of the approaches in this competition. |
| Data exclusions | For training and testing datasets, inclusions and exclusion criteria were described in Methods.<br><br>For overall participation and performance analysis, all valid submissions were included. For subsequence class-specific analysis, as the size of the test dataset is large, many teams chose to submit either to public leaderboard (for method development and validation) or private leaderboard (for final ranking) only. Higher percentage of missing scores was observed in lower ranking team, therefore, all teams outside top 50 were excluded from the performance analysis. |
| Replication | As this is a code competition, all code submission for inference were collected and graded automatically, which allows for reproduction of the scores. Final ranking was determined after re-runing all team's chosen submissions.<br><br>All training data and HPAv20 are publicly available for model training and testing dataset is available on Kaggle for replication of the performance. Teams independently make their code base open-sourced on Github, especially for the top 4 teams. |
| Randomization | Assembling the test set was based on criteria described in Methods, but the individual images were sampled randomly. Annotation of cells in the test set was done individually, in random order and assigned randomly to annotators to avoid bias. |
| Blinding | The participants did not have access to any cell labels in all datasets as well as images from the private test set, ensuring a fair evaluation. As with code competition, scoring algorithm was performed automatically without manual intervention. |

# Reporting for specific materials, systems and methods

We require information from authors about some types of materials, experimental systems and methods used in many studies. Here, indicate whether each material, system or method listed is relevant to your study. If you are not sure if a list item applies to your research, read the appropriate section before selecting a response.

## Materials & experimental systems

| n/a | Involved in the study |
|---|---|
| ☒ | Antibodies |
| ☒ | Eukaryotic cell lines |
| ☒ | Palaeontology and archaeology |
| ☒ | Animals and other organisms |
| ☒ | Human research participants |
| ☒ | Clinical data |
| ☒ | Dual use research of concern |

## Methods

| n/a | Involved in the study |
|---|---|
| ☒ | ChIP-seq |
| ☒ | Flow cytometry |
| ☒ | MRI-based neuroimaging |

