## [Peer Review File · Nature Methods]

Peer Review Information

Manuscript Title: Analysis of the Human Protein Atlas Weakly-Supervised Single Cell Classification Competition

Corresponding author name(s): Emma Lundberg

Reviewer Comments & Decisions:

Decision Letter, initial version:

Dear Emma,

Your Stage 1 Registered Report entitled "Analysis of the Human Protein Atlas Weakly-Supervised Single Cell Classification Competition" has now been seen by three reviewers, whose comments are attached. While they find your work of potential interest, they have raised serious concerns which in our view are sufficiently important that they preclude publication of the work in Nature Methods, at least in its present form.

As you will see, the reviewers have a lot of suggestions for the study, some of which (especially from referee 1) raise fundamental concerns about the design, which of course cannot be addressed since the competition is already underway.

Although we think these concerns are substantial enough to reject the paper, if you think you could better explain your design decisions in such a way that would convince the referees of the validity and benefits of the competition, we would be happy to consider such points.

We are also interested in hearing your thoughts on the additional concerns and suggestions raised by the referees.

If you are interested in revising this manuscript for submission to Nature Methods in the future, please contact me with a point-by-point rebuttal and to discuss your appeal before making any revisions. Otherwise, we hope that you find the reviewers' comments helpful when preparing your ultimate paper for submission elsewhere.

Sincerely,
Rita

Rita Strack, Ph.D.
Senior Editor
Nature Methods

Reviewers' Comments:

Reviewer #1:

Remarks to the Author:

The paper presents a methodology to analyze the data of the Kaggle competition run from Jan to May 2021 for single-cell classification of protein localization patterns. According to the manuscript, the task of the competition is to simultaneously produce segmentation masks and protein localization labels for single cells. Although this task seems technically correct in terms of computer vision methods, the biological motivation for the design of this task is not clearly stated. Training models that achieve this goal with only image-based labels (weakly supervised learning, without ground truth segmentations or single cell labels) seems unnecessarily complex from the technical point of view, especially considering that cell segmentation can be approached independently. Therefore, it is not clear how the innovations of the proposed competition will advance biological image analysis. The manuscript needs significant improvements before resulting in a meaningful contribution to the imaging community. See detailed comments below.

The challenge combines two problems

It appears that the interesting biological question behind this study is how to recognize protein localization patterns in single cells. However, due to a lack of ground truth annotations for both single cell classification and segmentation, the organizers challenge the community to solve both problems simultaneously. By combining those two challenges instead of disentangling them, the fundamental biological question gets lost in the computational details and assumptions. The authors seem to have the expectation that CAM-like methods can be used to segment the cells given a classification model, and the manuscript reads as if they are expecting these types of solutions from the participants of the competition. But, from an image processing perspective, cell segmentation can usually be achieved independently without any protein labels (fluorescent stains or localization classes). In the opposite direction, it is clear that accurate segmentation affects the quality of single cell classification: two cells that are not correctly separated by the segmentation model may have different classification labels, resulting in output errors for both cells in the protein localization task. The authors need to clarify their

goal, specifically, whether they want to improve segmentation or classification with the provided weakly annotated data.

Weakly supervised object detection and segmentation.

In computer vision literature, segmenting objects in a weakly supervised fashion is an active area of research. However, in contrast to the proposed challenge, weakly supervised object segmentation assumes that objects in images have one class that defines their type and spatial boundaries (dog, cat, car, etc). In the proposed challenge, all objects are cells, and the label does not define the type of object, i.e., the class label only defines the inner structure of the object but does not change the fact that all of them are cells. A more appropriate analogy would be to localize faces in photographs given the labels of their emotional expressions (sad, happy, angry). In this case, running an independent face detector first and then classifying the emotions is likely to be more successful than trying to learn where faces are located given their emotional labels. The combined challenge of segmenting and classifying cells would make sense if instead of segmenting the entire cell, the authors expect the methods to segment the subregion of the cell that displays the corresponding protein localization pattern (or the subregion of the face that hints the emotion).

Cell segmentation

Cell segmentation is a challenging problem on its own, independently of the protein localization label of cells, and usually approached in a fully supervised regime. Recent developments for cell segmentation could be leveraged for aiding the single cell classification problem proposed in this challenge by first using existing pre-trained models such as NucleAIzer, Cellpose, and Mesmer. These methods are not mentioned or considered in this work either as baselines (for their own segmentation challenge) or as potential pre-processing steps in the pipeline. It is unclear if the segmentation model described in line 178 was trained with manually generated annotations or with automatic annotations from other tools. Furthermore, if the segmentations used in the competition for evaluation are generated with automatic segmentation methods, it cannot be considered ground truth segmentation. The manuscript needs to clarify how these annotations were obtained. A deep learning model was used for segmentation of nuclei, but the cell body was segmented using classical algorithms (watershed). Are these masks used for evaluating participants of the competition? This may have biases that result in unfair evaluation. Are these masks available on the entire dataset (training / validation)? The description of these details could be improved in the manuscript.

Single cell classification

The manuscript indicates that segmentation is used as an auxiliary task for obtaining class labels, but it is unclear how segmentation would support protein localization classification. Given that cell segmentation can be achieved without the protein channel, there is a disconnect between the goal of using weak labels for segmenting cells and simultaneously guessing the right localization pattern.

Instead of being an auxiliary task, segmentation is a required preliminary step for classification. Under this definition, classifying single cells would be the actual challenge, because all the independent advances made in cell segmentation during the last few years could be used in preparation for classification. Therefore, it is unclear what would be the technical challenge of classifying single cells after they have been independently segmented, i.e., existing classification approaches can be readily used.

According to previous research conducted by the authors, protein localization is defined as a multi-label classification problem and they assert the same property holds for single cells. The authors describe their plan to compare single cell classification with image-based classification obtained in the previous challenge, but it is not clear in the manuscript how these results could be compared and analyzed. This is perhaps the most interesting question and closely related to biological insights. The problem is not how to simultaneously segment and assign labels to single cells, the problem is how to assign labels to pre-segmented cells when they have not been annotated individually. Framing the challenge in this way would bring interesting methods and would help answer biological questions related to protein localization patterns that have not been studied before at single cell resolution.

Reviewer #2:

Remarks to the Author:

This paper will describe the results of an ongoing Kaggle competition, which aims at generating single cell level protein localization labels from training set annotated at the image level, in order to place the competitors in a weakly-supervised setting, alleviating the burden of obtaining hand-labeled data sets. The weakly-supervised aspect of this multi-label classification problem relies on 1) annotation not directly giving the expected solution for this task (imprecise label), because the class are given at the full image level and with no single cell identification and 2) the use of existing resources (previous competition ref [14]).

Compared to the first competition which was at the image level, the purpose is to target single-cell variations with task being to perform segmentation (of cells only, see below?) and classification.

Another main difference is that the code to apply a model will all be run from jupyter notebooks, allowing a fair assessment of code performance and giving constraint to the application of the models, which will be trained independantly. The challenge is a big add on to the previous HPA image challenge, now focusing on single cells, and the analysis of results is expected to follow the same lines as the first paper (ref 14) with some adjustments regarding cell segmentation which may be inherent to some solutions, and new interpretation of the atlas allowed by the developed methods which will allow to unprecedentedly augment the quantity of available information in the cell atlas important resource, and open exciting possibilities when shared. The authors proposed for example to explore the applicability of the trained models to study cell cycle dependant variations which will be indeed of great interest.

The analysis of results will give the performance of pattern class identification, but also of the auxiliary task of cell mask segmentation. It will also try to analyse in depth the different best performing models with visual maps of class activation CAM (only for cnn-based solutions) and visualisation of subcellular clusters. An ablation study is also aimed to be performed.

Because of the auxiliary task of cell segmentation, and of the introduction enhancing the importance of quantification of expression in addition to their spatial patterns, I found that a clear introduction was missing regarding the expected task.

It is still unclear for me, in particular regarding the introduction also emphasizing the fact that one protein can have several localization in one cell and also that these localization can be unbalanced, if the participants task was to either identify single cells, and give the list of localization label (with no position in the image apart) to the protein of interest OR to identify single cells, and to perform instance segmentation of proteins (i.e pixel segmentation and spatial patterns class). This may come from the lack of figures at this stage, but some text edition could also help to make it clearer.

It is mentioned that single-cells level label with all labels corresponding to image-labels were removed, I could not figure out why by my self, could you elaborate? Also in your previous paper [14] it was stated : "However, we expect the impact to be minimal because only ~2% of proteins vary in their localization patterns between cells in images" : then how many proteins were selected following this rule of imposing image with cells showing variability?

Because this challenge is built upon the previous one [ref 14] , with annotation sand training data sets still available, it would be of interest to create a figure that would clearly compare the task in both competitions, and the training data set level of annotations. In particular because the previous dataset is released and made available to the participants to train their model, it is important to underline how different is the task so it can still be considered as weakly supervised, in particular because most of the image from the first competition was presenting uniform cell class (meaning one type of localization/label) , so it could be used directly as additional training data set by simply segmenting the cells, so the challenge here would have been not that weakly supervised. Could you comment on this point to justify the usage of weakly supervised?

The dataset represents 17 different cell lines , and 19 class including the negative one are considered, while the first challenge considered 28 labels and 27 different cell lines. The explanation about the reduction of class (getting a simplified balanced dataset) comes very late in the method, but there is no explanations about the reduction in cell lines. Please add a few words about this choice in the presentation of the challenge.

In particular in the methods it is specified that 30 labels were used for the annotation process, and the grouping in 19 class is summarized as "functionally and spatially similar", giving the list of grouped class. For example Focal adhesions (FAs) and Actin filaments has been grouped as Actin Filament: this is for example a choice that could be discussed because FAs have usually a very different shape than filament, and I do not think they can be considered to have the same function.

Ablation studies are of interest to try to understand better the causality of the obtained results, however in the previous study [14] the results were underexploited. Maybe focusing the ablation study on the input data rather than changing the architecture of the loss function could lead to a better interpretation of the robustness of the different models and their applicability for the readers.

Super minor:

The proportion /list of labels was not given (to detect "rare" class).

While domain knowledge is interesting to be assessed among the participants, it is not clear what is the definition of domain here, and this should be defined in the question asked to the participants.

Reviewer #3:

Remarks to the Author:

The authors propose a competition to evaluate cell segmentation and classification methods for the purpose of detecting, localizing, and identifying protein patterns in fluorescence microscopy images displaying multiple cells. The competition is open to the community and aims to spur the development of novel segmentation and classification approaches capable of dealing with the limited availability of annotations at the single-cell level.

This competition is timely, as single-cell analyses in imaging (as well as in other technologies, such as scRNA-seq technologies) become prevalent. The classification task is certainly challenging in that methods will have to deconvolve multiple image-level labels onto high resolution single-cell predictions. The fact that the competition will allow the use of additional sources of data beyond those provided by the organizers is aligned with current trends in machine learning positing that both data and methods are key for performance breakthroughs.

One comment is related to the fact that cell phenotypes are notoriously hard to bin onto well defined categories, mostly because one observes a continuum of phenotypes as opposed to discrete patterns (Liberali, P., Snijder, B. & Pelkmans, L. Single-cell and multivariate approaches in genetic perturbation screens. *Nat Rev Genet* 16, 18–32 (2015)). While a classification task certainly makes it easier to evaluate different methods, the authors are encouraged to provide a discussion on other end-points for evaluation (e.g., phenotypic distances among protein pattern populations) that would better translate to the novelty-seeking scenarios where most biology studies are undertaken.

The evaluation criterion for the proposed study merges the performance of both segmentation and classification tasks. This is fine but it would be good to have in addition more details on the segmentation task alone. In that regard, there is already an on-going cell segmentation benchmarking effort (cf. <http://celltrackingchallenge.net/latest-csb-results/>; see also Ulman, V., Maška, M., Magnusson, K. et al. An objective comparison of cell-tracking algorithms. *Nat Methods* 14, 1141–1152 (2017)) to which the current study should better align in terms of terminology (e.g., the IoU metric is the Jaccard index in the cell segmentation benchmark) as well as metrics and ranking approach.

Another issue is related to the fact that images where all cells have the same label as the label associated with the entire image are removed (i.e., images with entirely homogeneous populations). These images are actually a baseline case that the methods should be able to handle as well, so it's not clear why these images are being removed. In general, it's not too clear to me why the study is focusing only on pronounced heterogeneity.

A more specific question relates to the private and public test sets. It would be good to disclose (either before or after the competition) the degree of overlap (in terms of image or phenotypic similarity) between them.

One suggestion for the analysis is to investigate whether cell-to-cell

interactions (c.f. Bechtel, T.J., Reyes-Robles, T., Fadeyi, O.O. et al. Strategies for monitoring cell–cell interactions. *Nat Chem Biol* 17, 641–652 (2021)) or agglomerations are informative with regards to heterogeneity and pattern distributions as well as to method performance. This should be possible to analyze as single-cell locations and outlines would be provided by the methods.

Author Rebuttal to Initial comments

RESPONSE TO REVIEWER COMMENTS

Reviewer #1:

Remarks to the Author:

The paper presents a methodology to analyze the data of the Kaggle competition run from Jan to May 2021 for single-cell classification of protein localization patterns. According to the manuscript, the task of the competition is to simultaneously produce segmentation masks and protein localization labels for single cells. Although this task seems technically correct in terms of computer vision methods, the biological motivation for the design of this task is not clearly stated. Training models that achieve this goal with only image-based labels (weakly supervised learning, without ground truth segmentations or single cell labels) seems unnecessarily complex from the technical point of view, especially considering that cell segmentation can be approached independently. Therefore, it is not clear how the innovations of the proposed competition will advance biological image analysis. The manuscript needs significant improvements before resulting in a meaningful contribution to the imaging community. See detailed comments below.

We thank the reviewer for his/her extensive and constructive comments to our registered report.

There seems to be a misunderstanding of what the actual task of the challenge is and a misconception of how it was designed with regards to the segmentation and classification tasks. We will make sure this is fully clear in the full length manuscript, and include a Figure to illustrate the challenge design. Please see further clarifications in the point-by-point answers to the comments below.

We believe that innovations of the proposed competition will advance biological image analysis in several different regards, and will emphasize these in the full manuscript. In the era of single cell analysis, with ever-increasing single-cell omics data generation and advances in techniques to distill knowledge from sequencing reads, we lack techniques to extract spatial features of protein localization at single-cell resolution in images. There are plenty of image-level classification models available, such as those in our previous competition (Ouyang *et al.* “Analysis of the Human Protein Atlas Image

Classification competition". Nat Methods 16, 1254–1261 (2019). <https://doi.org/10.1038/s41592-019-0658-6>), but single-cell models are under-developed. We agree with the reviewer that the segmentation can be approached separately (see answers to comments below), but see two key arguments for the weakly-supervised challenge setting. First, even with well-segmented cell masks, obtaining the amount of ground-truth labels for individual cells for fully supervised models is too costly to implement in practice. Second, providing a weakly-labelled dataset will help to draw attention from experts from the AI field and use it as a benchmark dataset for development of new weakly-supervised algorithms. Therefore we decided to design the challenge to encourage innovation in making use of image-level datasets for precise cell level classification. We envision that a finer mapping of protein distribution would be useful in making novel discoveries in biology, importantly, without the extra efforts of providing single-cell ground-truth labels for training. Since no single-cell groundtruth labels are provided for training models, it's also natural to not provide already segmented cells, and make the challenge setting closer to what we have in reality.

Having received feedback from the participants, we are convinced that the challenge design was unique, challenging and fun; and inspired (unusually) many of the top Kagglers to participate; in other words it was difficult, but not too complex. With the results of the competition in hand, we are also convinced that the developed models will make a significant and immediate contribution to the biological imaging community, and for bringing quantitative spatial omics to the single cell level.

The challenge combines two problems.

It appears that the interesting biological question behind this study is how to recognize protein localization patterns in single cells. However, due to a lack of ground truth annotations for both single cell classification and segmentation, the organizers challenge the community to solve both problems simultaneously. By combining those two challenges instead of disentangling them, the fundamental biological question gets lost in the computational details and assumptions. The authors seem to have the expectation that CAM-like methods can be used to segment the cells given a classification model, and the manuscript reads as if they are expecting these types of solutions from the participants of the competition. But, from an image processing perspective, cell segmentation can usually be achieved independently without any protein labels (fluorescent stains or localization classes). In the opposite direction, it is clear that accurate segmentation affects the quality of single cell classification: two cells that are not correctly separated by the segmentation model may have different classification labels, resulting in output errors for both cells in the protein localization task. The authors need to clarify their goal, specifically, whether they want to improve segmentation or classification with the provided weakly annotated data.

We respectfully disagree with the reviewer opinion that the challenge would have been more successful given a design strictly separating the task of segmentation and classification, and will explain why below.

Again, the apparent misconception of the challenge task likely contributes to this comment, so please allow us to clarify this (will also be clarified in the manuscript) with the help of Figure 1 below.

A: Challenge Data Setup

B: Example Approaches

Figure 1. Challenge overview. A. Dataset setup for the competition. Labelled images were available for the competitors to use. The test sets consisted of individually labelled cells and were kept hidden from the competitors. The public testset is used by the participants to test their models for a ranking on the live leaderboard, while the private testset is used for the final ranking. B. With the design of the challenge, we envision different types of approaches of single cell classification from weak-labels being used with two examples illustrated here: The ‘Sequential segmentation and classification’ approach where single cells are first segmented before patterns are classified for each single cell, and the “Parallel segmentation and activation maps” approach where the entire image is simultaneously analyzed using Class Activation Mapping and the resulting maps are combined with separately segmented cells for final single cell classification.

The main challenge of this competition is single cell classification on a weakly labeled training dataset (Figure 1A). The segmentation of each cell is required to identify which cell corresponds to which label(s). The participants had the possibility (but were not required) to make segmentation as an

auxiliary task to the classification model, which could potentially improve both the segmentation and classification results, and make the solution more generalizable by understanding the context from which the labels are predicted. Instead of closing the competition in a mode with segmentation followed by classification, **we deliberately chose this open design to allow participants the flexibility to design their solution to solve both problems simultaneously or separately.**

From a deep learning perspective, weakly-supervised classification is an area of much needed progress because of the expensive and non-scalable process of manually annotating single objects in large datasets, which applies to single cell research. For weakly-supervised challenges, it is common to have a segmentation mask to identify the instant. The goal of this challenge is to do single-cell classification, with segmentation as a way to identify which cell has what label. Cells are matched with ground-truth at $\text{IOU} \geq 0.6$, which is more than enough to identify separate cells.

To simplify for the participants that want to focus on the classification task only, we provided them with our segmentation model (HPACellSegmentator), aligning with 90% of the groundtruth single cell masks with passing IoU threshold of 0.6 (see more details in the response to the cell segmentation comment below). We also mentioned to the participants other methods for segmentation, like Cellpose. We will clarify this point in the manuscript. Based on this we believe it was very clear to the participants that they could approach the segmentation and classification problem as separate, or as one.

We would also like to clarify that we did not expect CAM-like approaches for segmentation. We did however expect CAM-based approaches for the classification task to be prevalent because in weakly-supervised research, CAM-based methods are state-of-the-art (Figure 1B). Based on the results from the competition, we can now see that CAM-based approach was indeed one of the most popular approaches. More than half of the top 30 teams utilized CAM-based approaches, including the 1st and 3rd winners. (Table 1). The successful results of the now closed competition also attests to the fact that the participants were not confused as to what the actual task was, and that by having an open design (i.e. and not limiting the tasks in 2 steps), we benefited from a highly diverse group of solutions (Table 1)

Team	Pseudo-labeling	Manual labeling	Cell culling	Cell Tiling	Cell sampling	16 bit input	Edge Heuristics	Image & Cell Models	Multi-task	CAM/Attention	Transformer model	Previous challenge model	Weighted loss / Oversampling	Non-BCE loss
1	X				X		X	X	X	X	X			X
2					X		X	X	X				X	
3	X	X			X	X		X		X			X	X
4	X	X	X				X							X
5	X	X	X	X						X			X	
6				X						X	X		X	
7	X						X	X			X			X
8					X		X			X				
9					X			X						
10			X				X	X						

Definitions:

Pseudo-labeling: Predict labels for unlabeled data in order to increase the size of the dataset.
 Manual labeling: Manual labeling of rarer classes, to increase training-set size.
 Cell culling: Some heuristic for removing "bad" cells from the training data was used.
 Cell tiling: Assemble crops of cells into full images, using the cropped cells as "tiles".
 Cell sampling: Sample individual cells from images.
 16 bit input: 16-bit images were used as input in some part of the model.
 Edge heuristics: Some heuristics for removing cells from the edges of images
 Image & Cell Models: Separate models for images and cells
 Multi-task: Train models using multiple simultaneous tasks
 CAM/Attention: At least part of the solution included a CAM or an Attention layer.
 Transformer model: The usage of transformer model
 Previous challenge model: Usage of models from the previous HPA Kaggle Challenge
 Weighted loss / Oversampling: Usage of either weighted loss and/or oversampling
 Non-BCE loss: Usage of other loss functions than Binary Cross Entropy

Table 1: Overview of different approaches used by the top 10 teams in the competition.

Weakly supervised object detection and segmentation.

In computer vision literature, segmenting objects in a weakly supervised fashion is an active area of research. However, in contrast to the proposed challenge, weakly supervised object segmentation assumes that objects in images have one class that defines their type and spatial boundaries (dog, cat, car, etc). In the proposed challenge, all objects are cells, and the label does not define the type of object,

i.e., the class label only defines the inner structure of the object but does not change the fact that all of them are cells.

A more appropriate analogy would be to localize faces in photographs given the labels of their emotional expressions (sad, happy, angry).

In this case, running an independent face detector first and then classifying the emotions is likely to be more successful than trying to learn where faces are located given their emotional labels.

The combined challenge of segmenting and classifying cells would make sense if instead of segmenting the entire cell, the authors expect the methods to segment the subregion of the cell that displays the corresponding protein localization pattern (or the subregion of the face that hints the emotion).

As discussed above, the main task of this challenge is weakly-supervised classification, with cell segmentation as a way of identifying the cell whose label belongs to. The participants can choose to make cell segmentation an auxiliary task or not. The proposed method of first running a face detector and then classifying the emotions is analogous to running the cell segmentation model provided by us to the participants followed by a classification algorithm which is the main task of the challenge. Again, we will ensure to clarify this in the full manuscript text.

One important reason not to strictly define the solution to be a 2-step procedure is the fact that we don't know which is more efficient (separate or combined tasks), and therefore we kept an open design (see answer to the comment above). We see no advantage in fixing the format of the competition or solution design as it would limit the novelty of participants' approaches. There are studies in the literature where solving both classification and segmentation together achieved good results. For example, the authors used maskRCNN to do both segmentation and classification of cervical cells in pap smear images (Kurnianingsih *et al.*, "Segmentation and Classification of Cervical Cells Using Deep Learning," *IEEE Access*, vol. 7, pp. 116925-116941, 2019, doi: 10.1109/ACCESS.2019.2936017.) or breast tumors in sonogram (Chiao, Jui-Ying *et al.* "Detection and classification the breast tumors using mask R-CNN on sonograms." *Medicine* vol. 98,19 (2019): e15200. doi:10.1097/MD.00000000000015200).

Although we would love to see robust organelle segmentation models (i.e. subregions of cells), as proposed, we believe that this would be too complex as a challenge given the current state-of-the art for organelle segmentation and the fact that 50% of all proteins are found in multiple organelles. We consider this outside of the scope of what kind of competition we could realistically host at the moment, however such a competition may turn out feasible in the future, and we will mention it in the discussion.

Cell segmentation

Cell segmentation is a challenging problem on its own, independently of the protein localization label of cells, and usually approached in a fully supervised regime. Recent developments for cell segmentation could be leveraged for aiding the single cell classification problem proposed in this challenge by first using existing pre-trained models such as NucleAIzer, Cellpose, and Mesmer. These methods are not

mentioned or considered in this work either as baselines (for their own segmentation challenge) or as potential pre-processing steps in the pipeline. It is unclear if the segmentation model described in line 178 was trained with manually generated annotations or with automatic annotations from other tools. Furthermore, if the segmentations used in the competition for evaluation are generated with automatic segmentation methods, it cannot be considered ground truth segmentation. The manuscript needs to clarify how these annotations were obtained. A deep learning model was used for segmentation of nuclei, but the cell body was segmented using classical algorithms (watershed). Are these masks used for evaluating participants of the competition? This may have biases that result in unfair evaluation. Are these masks available on the entire dataset (training / validation)? The description of these details could be improved in the manuscript.

We thank the reviewer for raising these concerns about segmentations and will clarify the details in our manuscript. The segmentation ground truth for the test set is manually curated by annotators, so it's not biased to any model. In specific, the base for the ground-truth segmentation of the test set, as described in the Methods, was generated by HPACellSegmentator, which predicted both nuclei and cell body, while watershed was only used for a small step in the post-processing. After that, each cell was checked and adjusted manually by our expert annotators. Furthermore, the scoring metrics considers matching cells with IOU of 0.6, thereby not penalizing imperfect segmentation.

We agree that cell segmentation is a challenging problem on its own. We encouraged participants to use any types of cell segmentation that they can possibly find to be suitable, and in fact specifically mentioned Cellpose in the "Welcome message" for the competition (<https://www.kaggle.com/c/hpa-single-cell-image-classification/discussion/214616>). We provided HPACellSegmentator because it was trained on hand-annotated segmentations of HPA data. We can now see that most participants did use the HPACellSegmentator or an adjusted version of the HPACellSegmentator (participants modified it for speed, post processing etc). In the full manuscript, we will review and analyze the different segmentation approaches used by the participants.

Single cell classification

The manuscript indicates that segmentation is used as an auxiliary task for obtaining class labels, but it is unclear how segmentation would support protein localization classification. Given that cell segmentation can be achieved without the protein channel, there is a disconnect between the goal of using weak labels for segmenting cells and simultaneously guessing the right localization pattern. Instead of being an auxiliary task, segmentation is a required preliminary step for classification. Under this definition, classifying single cells would be the actual challenge, because all the independent advances made in cell segmentation during the last few years could be used in preparation for classification. Therefore, it is unclear what would be the technical challenge of classifying single cells after they have been independently segmented, i.e., existing classification approaches can be readily used.

As for the technical challenges for classifying single cells, one thing to keep in mind is that there are no single-cell level ground-truth labels given to the participants for training (precise single cell labels are only available for tests which are invisible to the participants). Essentially, that makes the task completely different from classical fully-supervised learning tasks, and most of the existing methods are not likely to work. As of now, training weakly supervised models is one of the hot topics in the AI research community, and thus it remains challenging to train single cell image classification models without the exact ground truth labels, even if the participants are given segmented cells. Moreover, our datasets are also challenging in terms of high class imbalance and a multilabel setting, i.e. one cell can correspond to many labels which is also not typical for those commonly used reference dataset (e.g. ImageNet). In fact, most of our participants find the task is particularly challenging despite the fact that many of them are “Grandmasters” on Kaggle, the highest ranking members on the site.

Regarding how segmentation would support protein localization classification, we initially thought that it would be beneficial to combine the two tasks giving the success of joint prediction models such as MaskRCNN. We did not expect the segmentation will help the protein localization. However, we thought that the single cell classification task might help improve the segmentation results (inspired by MaskRCNN) giving the additional information from the protein channel, especially for cases where cells are crowded and lack a clear separation from the reference channels alone. Now that the challenge has been closed, we conclude that maskRCNN approaches were used at the beginning of competition but then teams quickly moved to CAM-based approaches and used our segmentation model independently as the reviewer also mentioned.

Nevertheless, our challenge setting embraces the two possibilities, and we do see some participants (including the 1st place winner) trying to improve the segmentation step and improve their classification score, which matches well with our expectation. We will evaluate and discuss the different approaches in these regards in the manuscript.

According to previous research conducted by the authors, protein localization is defined as a multi-label classification problem and they assert the same property holds for single cells. The authors describe their plan to compare single cell classification with image-based classification obtained in the previous challenge, but it is not clear in the manuscript how these results could be compared and analyzed. This is perhaps the most interesting question and closely related to biological insights.

We agree that this is highly interesting and therefore proposed such a comparison as part of the analysis plans for the manuscript. In particular, we will compare this challenge’s winning model with the previous winner by comparing the F1 scores of the solutions, their feature maps (for example by using activation maps), and comparing feature cluster qualities.

The problem is not how to simultaneously segment and assign labels to single cells, the problem is how to assign labels to pre-segmented cells when they have not been annotated individually. Framing the challenge in this way would bring interesting methods and would help answer biological questions related to protein localization patterns that have not been studied before at single cell resolution.

Framing the challenge this way is one approach to solve the problem. As we described above, we kept an open design to encourage diversity and innovation in solutions, and will make sure to clarify this in the full manuscript. We thank the reviewer for seeing the value of this challenge in terms of novel interesting methods and possibilities to study previously intractable biological questions.

Reviewer #2:

Remarks to the Author:

This paper will describe the results of an ongoing Kaggle competition, which aims at generating single cell level protein localization labels from training set annotated at the image level, in order to place the competitors in a weakly-supervised setting, alleviating the burden of obtaining hand-labeled data sets. The weakly-supervised aspect of this multi-label classification problem relies on 1) annotation not directly giving the expected solution for this task (imprecise label), because the class are given at the full image level and with no single cell identification and 2) the use of existing resources (previous competition ref [14]).

Compared to the first competition which was at the image level, the purpose is to target single-cell variations with task being to perform segmentation (of cells only, see below?) and classification. Another main difference is that the code to apply a model will all be run from jupyter notebooks, allowing a fair assessment of code performance and giving constraint to the application of the models, which will be trained independantly. The challenge is a big add on to the previous HPA image challenge, now focusing on single cells, and the analysis of results is expected to follow the same lines as the first paper (ref 14) with some adjustments regarding cell segmentation which may be inherent to some solutions, and new interpretation of the atlas allowed by the developed methods which will allow to unprecedentedly augment the quantity of available information in the cell atlas important resource, and open exciting possibilities when shared. The authors proposed for example to explore the applicability of the trained models to study cell cycle dependant variations which will be indeed of great interest. The analysis of results will give the performance of pattern class identification, but also of the auxiliary task of cell mask segmentation. It will also try to analyse in depth the different best performing models with visual maps of class activation CAM (only for cnn-based solutions) and visualisation of subcellular clusters. An ablation study is also aimed to be performed.

We thank the reviewer for his/her constructive comments, and appreciate that he/she sees that this is a big add-on to the previous competition and sees both the need and benefit for the scientific community of the development of such models.

Because of the auxiliary task of cell segmentation, and of the introduction enhancing the importance of quantification of expression in addition to their spatial patterns, I found that a clear introduction was missing regarding the expected task.

We thank the reviewer for pointing this out. To clarify the task of the competition we will modify the introductory text and include a figure (Figure 1 below) to the full manuscript.

It is still unclear for me, in particular regarding the introduction also emphasizing the fact that one protein can have several localization in one cell and also that these localization can be unbalanced, if the participants task was to either identify single cells, and give the list of localization label (with no position in the image apart) to the protein of interest OR to identify single cells, and to perform instance segmentation of proteins (i.e pixel segmentation and spatial patterns class). This may come from the lack of figures at this stage, but some text edition could also help to make it clearer.

We agree with the reviewer that a schematic figure would immensely help to clarify this point and we will include such a Figure 1 (see below). In this competition, the participants are asked to identify single cells and give all localization labels for every cell.

The main challenge of this competition is single cell classification on a weakly labeled training dataset (Figure 1A). The segmentation of each cell is required to identify which cell corresponds to which label(s). The participants had the possibility (but were not required) to make segmentation as an auxiliary task to the classification model, which could potentially improve both the segmentation and classification results, and make the solution more generalizable by understanding the context from which the labels are predicted. Instead of closing the competition in a mode with segmentation followed by classification, **we deliberately chose this open design to allow participants the flexibility to design their solution to solve both problems simultaneously or separately.** This open design allows participants to be creative in their approaches (Figure 1B).

A: Challenge Data Setup

B: Example Approaches

Figure 1. Challenge overview. A. Dataset setup for the competition. Labeled images were available for the competitors to use. The test sets consisted of individually labelled cells and were kept hidden from the competitors. The public testset is used by the participants to test their models for a ranking on the live leaderboard, while the private testset is used for the final ranking. B. With the design of the challenge, we envision different types of approaches of single cell classification from weak-labels being used with two examples illustrated here: The ‘Sequential segmentation and classification’ approach where single cells are first segmented before patterns are classified for each single cell, and the “Parallel segmentation and activation maps” approach where the entire image is simultaneously analyzed using Class Activation Mapping and the resulting maps are combined with separately segmented cells for final single cell classification.

It is mentioned that single-cells level label with all labels corresponding to image-labels were removed, I could not figure out why by myself, could you elaborate? Also in your previous paper [14] it was stated :“However, we expect the impact to be minimal because only ~2% of proteins vary in their localization patterns between cells in images” : then how many proteins were selected following this rule of imposing image with cells showing variability?

There are several differences between the last competition and this one. One of the most important differences is that in the last challenge, we focused on aggregated labels of all cells in images, so we purposely removed most images with high single-cell variation (only 2% of the remaining images had single cell variations in this dataset). While in this challenge, we aim to study the heterogeneity of cells and proteins in the same images, therefore the test set was purposely chosen to have high single-cell variation within images. In total, about 20% of human proteins mapped in HPA show single cell variability (Mahdessian *et al.* "Spatiotemporal dissection of the cell cycle with single-cell proteogenomics". Nature 590, 649–654 (2021). <https://doi.org/10.1038/s41586-021-03232-9>). The confusion is understandable, and we will aim to make this point clear in the final paper.

Because this challenge is built upon the previous one [ref 14], with annotations and training data sets still available, it would be of interest to create a figure that would clearly compare the task in both competitions, and the training data set level of annotations. In particular because the previous dataset is released and made available to the participants to train their model, it is important to underline how different is the task so it can still be considered as weakly supervised, in particular because most of the image from the first competition was presenting uniform cell class (meaning one type of localization/label), so it could be used directly as additional training data set by simply segmenting the cells, so the challenge here would have been not that weakly supervised. Could you comment on this point to justify the usage of weakly supervised?

We thank the reviewer for this great suggestion. In the full manuscript we will include a supplementary Figure where we compare all aspects of the two competitions, including tasks, labels, datasets and metrics.

In the first challenge, the majority of training images were picked to present uniform cell classes (i.e. avoid single cell heterogeneity) because the task was focused on image-level classification. In the 2nd challenge, the data set (particularly the test sets) was sampled to have high single cell heterogeneity, which means that the individual cells will not inherit exactly the same classes as image-labels. To score high on the test set, the model needs to handle these image-level weak labels and predict cell-level labels.

Regarding the "weak labels" in our challenge, even through the training sample contains uniform distributed cells with known classes, our test set with high heterogeneity and particularly high cell-to-cell variability in spatial localisation, and we annotate each cell independently -- these make our training set a weakly labelled dataset in comparison. Essentially, to succeed in this challenge, a model needs to figure out the exact label for each cell and naively train models on segmented cells with image-level labels gives baseline results, but it is unlikely to succeed in the challenge. In fact, after the competition we indeed saw our challenge design stimulated a large variety of solutions, none of the winning solutions simply train models only on segmented cells with image-level labels.

Moreover, the naive solution might work for classes with many cell instances (e.g. Nucleoplasm) but it is unlikely to work for rare and high viability classes like Mitotic Spindle. For rare classes, you can't get enough training instances after the above mentioned exclusion. In fact we only saw low-ranking teams in this competition utilize this approach alone.

The dataset represents 17 different cell lines, and 19 class including the negative one are considered, while the first challenge considered 28 labels and 27 different cell lines. The explanation about the reduction of class (getting a simplified balanced dataset) comes very late in the method, but there is no explanations about the reduction in cell lines. Please add a few words about this choice in the presentation of the challenge.

We thank the reviewer for noticing this detail and will clarify this in the manuscript. Our single-cell annotated dataset is much smaller than the whole HPA. And there is an even greater imbalance for both cell lines and classes. So the decision to merge/exclude was to have a simplified and more balanced dataset in terms of both cell lines and classes. We have added the following text to the Methods explaining the reduction in cell lines:

"The final test set consisted of 1,776 images comprising 41,597 single cells from 17 human cell lines of varying morphology.... To match the test set, the training set was constructed by sampling images from the 17 cell lines above from the training set of the previous Kaggle challenge.... Furthermore, all publicly available HPA images of all cell lines were available as extra training data."

It's worth noting that not all cell lines are actively studied in the HPA, nor do they present the same extent of single cell heterogeneity. Hence we aimed to include the most important cell lines in this competition. Considering the difficulty of this challenge, compared to the previous one, we also took cell morphology into account, and removed some cell lines known to be visually different with common artifacts.

In particular in the methods it is specified that 30 labels were used for the annotation process, and the grouping in 19 class is summarized as "functionally and spatially similar", giving the list of grouped class. For example Focal adhesions (FAs) and Actin filaments has been grouped as Actin Filament: this is for example a choice that could be discussed because FAs have usually a very different shape than filament, and I do not think they can be considered to have the same function.

We agree with the reviewer that FAs and Actin filaments are indeed functionally different. The FA class was too small to include in itself, and since oftentimes proteins are localized to both FAs and actin filaments, and the fact that FAs are found at the tip of Actin filaments we reasoned that this was the most reasonable merger. We will update the manuscript text to avoid the misleading statement about similar functions.

Ablation studies are of interest to try to understand better the causality of the obtained results, however in the previous study [14] the results were underexploited. Maybe focusing the ablation study on the input data rather than changing the architecture of the loss function could lead to a better interpretation of the robustness of the different models and their applicability for the readers. We agree with the reviewer that ablation on input is helpful to test the robustness of models. We will focus the generalizability tests on other HPA datasets (more cell lines), but not focus on generalizability to other datasets for now. Interestingly, the winning teams have used some augmentations on the inputs and we also aim to investigate these input manipulations.

Super minor:

The proportion /list of labels was not given (to detect "rare" class).
The proportion of labels were purposely kept hidden because the competition was ongoing at that time. We will include all such details in the full manuscript.

While domain knowledge is interesting to be assessed among the participants, it is not clear what is the definition of domain here, and this should be defined in the question asked to the participants. In the post-competition survey, we defined domain knowledge as “knowledge in biology, cells and microscopy imaging”. After closing the competition and conducting the survey, we now know that 2 or the top 4 teams had domain knowledge in this competition, in contrast to in the previous competition where none of the top 5 teams did.

Reviewer #3:

Remarks to the Author:

The authors propose a competition to evaluate cell segmentation and classification methods for the purpose of detecting, localizing, and identifying protein patterns in fluorescence microscopy images displaying multiple cells. The competition is open to the community and aims to spur the development of novel segmentation and classification approaches capable of dealing with the limited availability of annotations at the single-cell level.

This competition is timely, as single-cell analyses in imaging (as well as in other technologies, such as scRNA-seq technologies) become prevalent. The classification task is certainly challenging in that methods will have to deconvolve multiple image-level labels onto high resolution single-cell predictions. The fact that the competition will allow the use of additional sources of data beyond those provided by the organizers is aligned with current trends in machine learning positing that both data and methods are key for performance breakthroughs.

We thank the reviewer for appreciating the challenge's motivation and the promise of single cell image classification models.

One comment is related to the fact that cell phenotypes are notoriously hard to bin onto well defined categories, mostly because one observes a continuum of phenotypes as opposed to discrete patterns (Liberali, P., Snijder, B. & Pelkmans, L. Single-cell and multivariate approaches in genetic perturbation screens. *Nat Rev Genet* 16, 18–32 (2015)). While a classification task certainly makes it easier to evaluate different methods, the authors are encouraged to provide a discussion on other end-points for evaluation (e.g., phenotypic distances among protein pattern populations) that would better translate to the novelty-seeking scenarios where most biology studies are undertaken.

It is true that cell phenotypes are often continuous, particularly when studying perturbed cells (which is not the case in this challenge). The data we have in the competition are cell lines in normal culture condition (log phase growth) and the cell phenotypes will for example represent a continuum of cell cycle states. With regard to the protein localization patterns, they can most often be categorized into well defined classes due the organellar compartmentalization of the cell (Thul *et al.* "A subcellular map of the human proteome". *Science* 356, eaal3321 (2017). <https://doi.org/10.1126/science.aal3321>). Nevertheless, due to the continuous transitions of cell states and protein localized to multiple compartments, the boundary between different patterns can be blurred.

We agree with the reviewer that the models developed in this competition can be proven useful in novelty-seeking scenarios. Even though the models have been trained on discrete labels, it is still possible to obtain a continuous representation of the localization patterns in feature space. In fact, by using UMAP visualization to show the learned features of the winning model from our last image-level classification competition (Ouyang *et al.* "Analysis of the Human Protein Atlas Image Classification competition". *Nat Methods* 16, 1254–1261 (2019). <https://doi.org/10.1038/s41592-019-0658-6>), we manage to identify nucleolar protein clusters corresponding to a novel dynamic subcompartment lining the rim of the nucleolus (Stenström *et al.* "Mapping the nucleolar proteome reveals a spatiotemporal organization related to intrinsic protein disorder". *Mol Syst Biol.* Aug;16(8):e9469 (2020). doi: 10.15252/msb.20209469. PMID: 32744794; PMCID: PMC7397901). Using this approach we will explore the HPA Cell Atlas for novel patterns at the single cell level using the models developed in this challenge.

The evaluation criterion for the proposed study merges the performance of both segmentation and classification tasks. This is fine but it would be good to have in addition more details on the segmentation task alone. In that regard, there is already an on-going cell segmentation benchmarking effort (cf. <http://celltrackingchallenge.net/latest-csb-results/>; see also Ulman, V., Maška, M., Magnusson, K. et al. An objective comparison of cell-tracking algorithms. *Nat Methods* 14, 1141–1152

(2017)) to which the current study should better align in terms of terminology (e.g., the IoU metric is the Jaccard index in the cell segmentation benchmark) as well as metrics and ranking approach.

The Kaggle platform does (unfortunately) not support double metrics. Therefore, we chose to use mAP, which is a popular metric for similar computer vision tasks, as classification metric (our main task) for all cells that passed the 0.6 IOU threshold. Since many of the winning teams adjusted our baseline segmentation model, we aim to compare the masks accuracy afterward using IOU, and in particular assess if the performance of the segmentation task correlates with performance of the classification task in the full manuscript.

Another issue is related to the fact that images where all cells have the same label as the label associated with the entire image are removed (i.e., images with entirely homogeneous populations). These images are actually a baseline case that the methods should be able to handle as well, so it's not clear why these images are being removed. In general, it's not too clear to me why the study is focusing only on pronounced heterogeneity.

The test set is chosen to be of pronounced heterogeneity to ensure that a model needs to figure out the exact label for each cell to be successful, while models naively trained on segmented cells with image-level labels will give baseline results are unlikely to succeed in this challenge setting. In fact, now when the competition is closed we can see that our challenge design stimulated a large variety of solutions, where none of the winning solutions simply train models only on segmented cells with image-level labels (See Table 1).

We assume that if the model can handle high heterogeneity images well then it will also be able to handle a more homogenous population. Inspired by this comment we plan to use unpublished HPA images with uniform label distribution to test this generalizability aspect in the full manuscript.

Team	Pseudo-labeling	Manual labeling	Cell culling	Cell Tiling	Cell sampling	16 bit input	Edge Heuristics	Image & Cell Models	Multi-task	CAM/Attention	Transformer model	Previous challenge model	Weighted loss / Oversampling	Non-BCE loss
1	X				X		X	X	X	X	X			X
2					X		X	X	X				X	
3	X	X			X	X		X		X			X	X
4	X	X	X				X							X
5	X	X	X	X						X			X	
6				X						X	X	X		
7	X						X	X		X				X
8					X		X			X				
9					X			X						
10			X				X	X						

Definitions:

Pseudo-labeling: Predict labels for unlabeled data in order to increase the size of the dataset.
 Manual labeling: Manual labeling of rarer classes, to increase training-set size.
 Cell culling: Some heuristic for removing “bad” cells from the training data was used.
 Cell tiling: Assemble crops of cells into full images, using the cropped cells as “tiles”.
 Cell sampling: Sample individual cells from images.
 16 bit input: 16-bit images were used as input in some part of the model.
 Edge heuristics: Some heuristics for removing cells from the edges of images
 Image & Cell Models: Separate models for images and cells
 Multi-task: Train models using multiple simultaneous tasks
 CAM/Attention: At least part of the solution included a CAM or an Attention layer.
 Transformer model: The usage of transformer model
 Previous challenge model: Usage of models from the previous HPA Kaggle Challenge
 Weighted loss / Oversampling: Usage of either weighted loss and/or oversampling
 Non-BCE loss: Usage of other loss functions than Binary Cross Entropy

Table 1: Overview of different approaches used by the top 10 teams in the competition.

A more specific question relates to the private and public test sets. It would be good to disclose (either before or after the competition) the degree of overlap (in terms of image or phenotypic similarity) between them.

We assume the reviewer did not mean “overlap” but instead “similarity” between images. There is **no overlap** between private and public test sets. Instead, they are similarly distributed in terms of labels

and cell lines. This is crucial to the success of challenges like this where participants rely on the public leaderboard for validation (unfortunately sometimes neglected, hence leading to major leaderboard shake-ups such as in recent bioimage competitions <https://www.kaggle.com/c/hubmap-kidney-segmentation/leaderboard> or <https://www.kaggle.com/c/vinbigdata-chest-xray-abnormalities-detection/leaderboard>). We were extremely careful when designing the challenge to ensure similar distributions in the different data sets, and the success can now be confirmed by the minimal shake-up in the final leaderboard (<https://www.kaggle.com/c/hpa-single-cell-image-classification/leaderboard>). We will supply a supplementary table/figure with all details about the distribution of public and private test sets in the full manuscript.

One suggestion for the analysis is to investigate whether cell-to-cell interactions (c.f. Bechtel, T.J., Reyes-Robles, T., Fadeyi, O.O. et al. Strategies for monitoring cell–cell interactions. Nat Chem Biol 17, 641–652 (2021)) or agglomerations are informative with regards to heterogeneity and pattern distributions as well as to method performance. This should be possible to analyze as single-cell locations and outlines would be provided by the methods.

We agree with the reviewer that there are many interesting aspects of cell-to-cell variability that can be explored given these models, one being which proteins change their location in relation to cell-cell interactions or cell crowding. Although these biological questions remain out of the scope of this Analysis Article, we like this suggestion and will include an analysis of model performance in sparse vs dense cell regions in the images.

Decision Letter, first revision:

Dear Emma,

Thanks again for your earlier note. Here is the formal decision email with link to resubmit as a standard Analysis.

Thank you for your letter asking us to reconsider our decision on your Analysis, "Analysis of the Human Protein Atlas Weakly-Supervised Single Cell Classification Competition". As I mentioned, we would be willing to consider the final version of your paper as an Analysis rather than a Registered Report.

* include a point-by-point response to our referees and to any editorial suggestions

- * address the points listed described below to conform to our open science requirements
- * ensure it complies with our general format requirements as set out in our guide to authors at www.nature.com/naturemethods
- * resubmit all the necessary files electronically by using the link below to access your home page

[Redacted] This URL links to your confidential home page and associated information about manuscripts you may have submitted, or that you are reviewing for us. If you wish to forward this email to co-authors, please delete the link to your homepage.

We hope to receive your revised paper within eight weeks. If you cannot send it within this time, please let us know. In this event, we will still be happy to reconsider your paper at a later date so long as nothing similar has been accepted for publication at Nature Methods or published elsewhere.

OPEN SCIENCE REQUIREMENTS

REPORTING SUMMARY AND EDITORIAL POLICY CHECKLISTS

When revising your manuscript, please submit reporting summary and editorial policy checklists.

DATA AVAILABILITY

Please include a “Data availability” subsection in the Online Methods. This section should inform readers about the availability of the data used to support the conclusions of your study, including accession codes to public repositories, references to source data that may be published alongside the paper, unique identifiers such as URLs to data repository entries, or data set DOIs, and any other statement about data availability. At a minimum, you should include the following statement: “The data that support the findings of this study are available from the corresponding author upon request”, describing which data is available upon request and mentioning any restrictions on availability. If DOIs are provided, please include these in the Reference list (authors, title, publisher (repository name), identifier, year). For more guidance on how to write this section please see:

<http://www.nature.com/authors/policies/data/data-availability-statements-data-citations.pdf>

CODE AVAILABILITY

Please include a “Code Availability” subsection in the Online Methods which details how your custom code is made available. Only in rare cases (where code is not central to the main conclusions of the paper) is the statement “available upon request” allowed (and reasons should be specified).

ORCID

Nature Methods is committed to improving transparency in authorship. As part of our efforts in this direction, we are now requesting that all authors identified as ‘corresponding author’ on published papers create and link their Open Researcher and Contributor Identifier (ORCID) with their account on the Manuscript Tracking System (MTS), prior to acceptance. This applies to primary research papers only. ORCID helps the scientific community achieve unambiguous attribution of all scholarly

contributions. You can create and link your ORCID from the home page of the MTS by clicking on 'Modify my Springer Nature account'. For more information please visit www.springernature.com/orcid.

Sincerely,
Rita

Rita Strack, Ph.D.
Senior Editor
Nature Methods

Author Rebuttal, first revision:

RESPONSE TO REVIEWER COMMENTS

Reviewer #1:

Remarks to the Author:

The paper presents a methodology to analyze the data of the Kaggle competition run from Jan to May 2021 for single-cell classification of protein localization patterns. According to the manuscript, the task of the competition is to simultaneously produce segmentation masks and protein localization labels for single cells. Although this task seems technically correct in terms of computer vision methods, the biological motivation for the design of this task is not clearly stated. Training models that achieve this goal with only image-based labels (weakly supervised learning, without ground truth segmentations or single cell labels) seems unnecessarily complex from the technical point of view, especially considering that cell segmentation can be approached independently. Therefore, it is not clear how the innovations of the proposed competition will advance biological image analysis. The manuscript needs significant improvements before resulting in a meaningful contribution to the imaging community. See detailed comments below.

We thank the reviewer for his/her extensive and constructive comments to our registered report.

There seems to be a misunderstanding of what the actual task of the challenge was and a misconception of how it was designed with regards to the segmentation and classification tasks. We have made the effort to explicitly clarify this in the full length manuscript, and included Figure 2 to illustrate the challenge design. Please see further clarifications in the point-by-point answers to the comments below.

We believe that innovations of the proposed competition will advance biological image analysis in several different regards, and have emphasized these in the full manuscript. In the era of single cell analysis, with ever-increasing single-cell omics data generation and advances in techniques to distill knowledge from sequencing reads, we lack techniques to extract spatial features of protein localization at single-cell resolution in images. There are plenty of image-level classification models available, such as those in our previous competition (Ouyang *et al.* “Analysis of the Human Protein Atlas Image Classification competition”. *Nat Methods* 16, 1254–1261 (2019). <https://doi.org/10.1038/s41592-019-0658-6>) and newer methods like Kobayashi *et al.* “Self-Supervised Deep-Learning Encodes High-Resolution Features of Protein Subcellular Localization” <https://doi.org/10.1101/2021.03.29.437595> used for analysis of the new OpenCell resource just published in *Science* (Cho *et al.* “OpenCell: Endogenous tagging for the cartography of human cellular organization”. *Science* 375, 6585 <https://www.science.org/doi/10.1126/science.abi6983>), but single-cell models are under-developed. We agree with the reviewer that the segmentation can be approached separately (see answers to comments below), but see two key arguments for the weakly-supervised challenge setting. First, even with well-segmented cell masks, obtaining the amount of ground-truth labels for individual cells for fully supervised models is too costly to implement in practice. Second, providing a weakly-labeled dataset will help to draw attention from experts from the AI field and use it as a benchmark dataset for development of new weakly-supervised algorithms. Therefore we decided to design the challenge to encourage innovation in making use of image-level datasets for precise cell level classification. We envision that a finer mapping of protein distribution would be useful in making novel discoveries in biology, importantly, without the extra efforts of providing single-cell ground-truth labels for training. Since no single-cell groundtruth labels are provided for training models, it's also natural to not provide already segmented cells, and make the challenge setting closer to what we have in reality.

Having received feedback from the participants, we are convinced that the challenge design was unique, challenging and fun; and inspired (unusually) many of the top Kagglers to participate; in other words it was difficult, but not too complex. With the results of the competition in hand, we are also convinced that the developed models will make a significant and immediate contribution to the biological imaging community, and for bringing quantitative spatial omics to the single cell level.

The challenge combines two problems.

It appears that the interesting biological question behind this study is how to recognize protein localization patterns in single cells. However, due to a lack of ground truth annotations for both single cell classification and segmentation, the organizers challenge the community to solve both problems simultaneously. By combining those two challenges instead of disentangling them, the fundamental biological question gets lost in the computational details and assumptions. The authors seem to have the expectation that CAM-like methods can be used to segment the cells given a classification model, and the manuscript reads as if they are expecting these types of solutions from the participants of the

competition. But, from an image processing perspective, cell segmentation can usually be achieved independently without any protein labels (fluorescent stains or localization classes). In the opposite direction, it is clear that accurate segmentation affects the quality of single cell classification: two cells that are not correctly separated by the segmentation model may have different classification labels, resulting in output errors for both cells in the protein localization task. The authors need to clarify their goal, specifically, whether they want to improve segmentation or classification with the provided weakly annotated data.

We respectfully disagree with the reviewer opinion that the challenge would have been more successful given a design strictly separating the task of segmentation and classification, and will explain why below. Again, the apparent misconception of the challenge task likely contributes to this comment, so please allow us to clarify this (which is also clarified in the manuscript) with the help of Figure 2 A,B below (Please see full Figure 2 in the main manuscript package).

Figure 2. Challenge overview. **A.** Dataset setup for the competition. For training, participants were provided access to a well balanced training set and the public HPA dataset, which consists of fluorescent images with four channels and corresponding image-level labels. The solutions' performance were assessed by a private dataset of 1776 images containing 41k single cells with corresponding single-cell labels. This dataset was divided into the public testset, which was used by the participants to test their models for a ranking on the live leaderboard, and the private testset, in which all images and their single-cell labels were completely hidden and used for the final ranking. **B.** Overview of main approaches: image-level models take in images and predict image-level labels, which can be combined with class activation maps of different classes and segmented cell regions to give the final cell-level labels. Image-level models can also be trained on whole images, but predict on bag-of-cell tiles at inference to get cell labels. Cell-level models take in single cell crops and predict single cell labels.

The main challenge of this competition is single cell classification on a weakly labeled training dataset (Figure 2A). The segmentation of each cell is required to identify which cell corresponds to which label(s). The participants had the possibility (but were not required) to make segmentation as an auxiliary task to the classification model, which could potentially improve both the segmentation and classification results, and make the solution more generalizable by understanding the context from which the labels are predicted. Instead of closing the competition in a mode with segmentation followed by classification, **we deliberately chose this open design to allow participants the flexibility to design their solution to solve both problems simultaneously or separately.**

From a deep learning perspective, weakly-supervised classification is an area of much needed progress because of the expensive and non-scalable process of manually annotating single objects in large datasets, which applies to single cell research. For weakly-supervised challenges, it is common to have a segmentation mask to identify the instant. The goal of this challenge is to do single-cell classification, with segmentation as a way to identify which cell has what label. Cells are matched with ground-truth at $\text{IOU} \geq 0.6$, which is more than enough to identify separate cells.

To simplify for the participants that want to focus on the classification task only, we provided them with our segmentation model (HPACellSegmentator), aligning with 90% of the groundtruth single cell masks with passing IoU threshold of 0.6 (see more details in the response to the cell segmentation comment below). We also mentioned to the participants other methods for segmentation, like Cellpose. We have clarified this point in the manuscript. Based on this we believe it was very clear to the participants that they could approach the segmentation and classification problem as separate, or as one.

We would also like to clarify that we did not expect CAM-like approaches for segmentation. We did however expect CAM-based approaches for the classification task to be prevalent because in weakly-supervised research, CAM-based methods are state-of-the-art (Figure 2B first pipeline). Based on the results from the competition, we can now see that CAM-based approach was indeed one of the most popular approaches. More than half of the top 30 teams utilized CAM-based approaches for either cell-level models or image-level models, including the 1st and 3rd winners. (Supplementary Table 2). The successful results of the now closed competition also attests to the fact that the participants were not confused as to what the actual task was, and that by having an open design (i.e. and not limiting the tasks in 2 steps), we benefited from a highly diverse group of solutions (Supplementary Table 2). Below is a concise summary of the solution overview from the top 10 teams:

Team	Rare label strategies		Segmentation strategies		Single cell strategies		Loss strategies	
	Pseudo-labeling	Manual labeling of rare class	Border cell classifier	Edge heuristics	Transformers	cell and image models	Weighed-loss / Oversampling	Combine 2+ losses
Team 1	X		X	X	X	X		X
Team 2			X	X		X	X	
Team 3	X	X				X	X	X
Team 4	X	X		X				
Team 5	X	X						
Team 6 *								
Team 7	X			X		X		
Team 8				X				
Team 9						X		
Team 10				X		X		

Definitions	
Pseudo-labeling:	Predict labels for unlabeled data to increase the size of the training set.
Manual labeling:	Manually label images containing rare classes, to increase the size of the training set.
Border cell classifier:	Separate classifier to remove non-whole cells from input images.
Edge heuristics:	Add heuristics to remove cells touching edges from input images.
Transformers:	The use of a transformer model for cell prediction
Cell and image models:	The usage of separate models for image and cell predictions.
Weighted loss / oversampling:	Using either weighted loss values or oversampling strategies for more efficient loss values.
Combine 2+ losses:	Using at least 2 different loss functions.

Table 1: Overview of different approaches used by the top 10 teams in the competition. *Team 6 did you use or did not supply information about these methods in their overview. For full summary, please refer to Supplementary Table 2.

Weakly supervised object detection and segmentation.

In computer vision literature, segmenting objects in a weakly supervised fashion is an active area of research. However, in contrast to the proposed challenge, weakly supervised object segmentation assumes that objects in images have one class that defines their type and spatial boundaries (dog, cat, car, etc). In the proposed challenge, all objects are cells, and the label does not define the type of object, i.e., the class label only defines the inner structure of the object but does not change the fact that all of them are cells.

A more appropriate analogy would be to localize faces in photographs given the labels of their emotional expressions (sad, happy, angry).

In this case, running an independent face detector first and then classifying the emotions is likely to be more successful than trying to learn where faces are located given their emotional labels.

The combined challenge of segmenting and classifying cells would make sense if instead of segmenting the entire cell, the authors expect the methods to segment the subregion of the cell that displays the corresponding protein localization pattern (or the subregion of the face that hints the emotion).

As discussed above, the main task of this challenge is weakly-supervised classification, with cell segmentation as a way of identifying the cell whose label belongs to. The participants can choose to make cell segmentation an auxiliary task or not. The proposed method of first running a face detector and then classifying the emotions is analogous to running the cell segmentation model provided by us to the participants followed by a classification algorithm which is the main task of the challenge. Again, we have clarified this in the full manuscript text.

One important reason not to strictly define the solution to be a 2-step procedure is the fact that we don't know which is more efficient (separate or combined tasks), and therefore we kept an open design (see answer to the comment above). We see no advantage in fixing the format of the competition or solution design as it would limit the novelty of participants' approaches. In addition, ground-truths for single cell segmentations and labels, especially generated by experts, on the whole training dataset and/or whole HPA are required to turn this into a classification-only task, which is laborious and unaffordable. There are studies in the literature where solving both classification and segmentation together achieved good results. For example, the authors used maskRCNN to do both segmentation and classification of cervical cells in pap smear images (Kurnianingsih *et al.*, "Segmentation and Classification of Cervical Cells Using Deep Learning," *IEEE Access*, vol. 7, pp. 116925-116941, 2019, doi: 10.1109/ACCESS.2019.2936017.) or breast tumors in sonogram (Chiao, Jui-Ying *et al.* "Detection and

classification the breast tumors using mask R-CNN on sonograms.” *Medicine* vol. 98,19 (2019): e15200. doi:10.1097/MD.00000000000015200).

Although we would love to see robust organelle segmentation models (i.e. subregions of cells), as proposed, we believe that this would be too complex as a challenge given the current state-of-the art for organelle segmentation and the fact that ~50% of all proteins are found in multiple organelles. We therefore consider this outside of the scope of what kind of competition we could realistically host at the moment, however such a competition may turn out feasible in the future.

Cell segmentation

Cell segmentation is a challenging problem on its own, independently of the protein localization label of cells, and usually approached in a fully supervised regime. Recent developments for cell segmentation could be leveraged for aiding the single cell classification problem proposed in this challenge by first using existing pre-trained models such as NucleAIzer, Cellpose, and Mesmer. These methods are not mentioned or considered in this work either as baselines (for their own segmentation challenge) or as potential pre-processing steps in the pipeline. It is unclear if the segmentation model described in line 178 was trained with manually generated annotations or with automatic annotations from other tools. Furthermore, if the segmentations used in the competition for evaluation are generated with automatic segmentation methods, it cannot be considered ground truth segmentation. The manuscript needs to clarify how these annotations were obtained. A deep learning model was used for segmentation of nuclei, but the cell body was segmented using classical algorithms (watershed). Are these masks used for evaluating participants of the competition? This may have biases that result in unfair evaluation. Are these masks available on the entire dataset (training / validation)? The description of these details could be improved in the manuscript.

We thank the reviewer for raising these concerns about segmentations and will clarify the details in our manuscript. The segmentation ground truth for the test set was initially generated by a DPN-Unet (a winning model of data science bowl 2018 <https://www.kaggle.com/c/data-science-bowl-2018>) which has reasonably good performance importantly, the generated masks were subsequently manually curated by annotators. In specific, the base for the ground-truth segmentation of the test set, as described in the Methods, was generated by HPACellSegmetator (a python library built with DPN-Unet), which predicted both nuclei and cell body, while watershed was only used for a small step in the post-processing. After that, each cell was checked and adjusted manually by our expert annotators. Furthermore, the scoring metrics considers matching cells with IOU of 0.6, thereby not penalizing imperfect segmentation.

As a baseline segmentation model, we prefer a model which can run inference faster even though it might not perform the best (keep in mind that it is a “kernel competition” on Kaggle where each submission has the same fixed execution time). Due to its simple post-processing in the DPN-Unet

(watershed), the inference speed is significantly higher than Cellpose which requires heavy post-processing. Since NucleAIzer relies on Matlab, its non-free license will likely hinder the participation of our competition. However, we didn't not compare DPN-Unet with Mesmer.

We agree that cell segmentation is a challenging problem on its own. We encouraged participants to use any types of cell segmentation that they can possibly find to be suitable, and in fact specifically mentioned Cellpose in the "Welcome message" for the competition (<https://www.kaggle.com/c/hpa-single-cell-image-classification/discussion/214616>). We provided HPACellSegmentator because it was trained on hand-annotated segmentations of HPA data. We can now see that most participants did use the HPACellSegmentator or an adjusted version of the HPACellSegmentator (participants modified it for speed, post processing etc). Because of this, in the full manuscript we noted our observation that there was no association between AP scores and segmentation masks.

Single cell classification

The manuscript indicates that segmentation is used as an auxiliary task for obtaining class labels, but it is unclear how segmentation would support protein localization classification. Given that cell segmentation can be achieved without the protein channel, there is a disconnect between the goal of using weak labels for segmenting cells and simultaneously guessing the right localization pattern. Instead of being an auxiliary task, segmentation is a required preliminary step for classification. Under this definition, classifying single cells would be the actual challenge, because all the independent advances made in cell segmentation during the last few years could be used in preparation for classification. Therefore, it is unclear what would be the technical challenge of classifying single cells after they have been independently segmented, i.e., existing classification approaches can be readily used.

As for the technical challenges for classifying single cells, one thing to keep in mind (besides the fact that there are no models for classification of protein patterns in single cells published to date, nor any published image datasets labeled at the single cell level that could be used for fully supervised training) is that there are no single-cell level ground-truth labels available to the participants for training (precise single cell labels are only available for tests which are invisible to the participants). Essentially, that makes the task completely different from classical fully-supervised learning tasks, and most of the existing methods are not likely to work. As of now, training weakly supervised models is one of the hot topics in the AI research community, and thus it remains challenging to train single cell image classification models without the exact ground truth labels, even if the participants are given segmented cells. Moreover, our datasets are also challenging in terms of high class imbalance and a multilabel setting, i.e. one cell can correspond to many labels which is also not typical for those commonly used reference dataset (e.g. ImageNet). In fact, most of our participants find the task is particularly challenging despite the fact that many of them are "Grandmasters" on Kaggle, the highest ranking members on the site.

Regarding how segmentation would support protein localization classification, we initially thought that it would be beneficial to combine the two tasks giving the success of joint prediction models such as MaskRCNN. We did not expect the segmentation will help the protein localization. However, we thought that the single cell classification task might help improve the segmentation results (inspired by MaskRCNN) giving the additional information from the protein channel, especially for cases where cells are crowded and lack a clear separation from the reference channels alone. Now that the challenge has been closed, we conclude that maskRCNN approaches were used at the beginning of competition but then teams quickly moved to CAM-based approaches and used our segmentation model independently as the reviewer also mentioned.

Nevertheless, our challenge setting embraces the two possibilities, and we do see some participants (including the 1st place winner) trying to improve the segmentation step and improve their classification score, which matches well with our expectation. We discussed these different approaches *in the “Overview of approaches” and “Strategies of the winning teams” sections, as well as in our supplementary notes.*

According to previous research conducted by the authors, protein localization is defined as a multi-label classification problem and they assert the same property holds for single cells. The authors describe their plan to compare single cell classification with image-based classification obtained in the previous challenge, but it is not clear in the manuscript how these results could be compared and analyzed. This is perhaps the most interesting question and closely related to biological insights.

We agree that this is highly interesting and therefore originally proposed such a comparison as part of the analysis plans for the manuscript. A first note is that as the producers of the HPA dataset since 15 years back, we know our data very well and know that the property of multi-localization holds true also for single cells. As the data is freely available and we even have a section on protein multi-localization (<https://www.proteinatlas.org/humanproteome/subcellular/multilocalizing>), this is something participants could look deeper into if they would like to.

As for the comparison, one can in a way consider this entire competition a comparison to the previous one, as the top ranking models and datasets are all available. For example, the previous winning approach was adopted by Team 12 in this competition, while Team 1 (also winner of the previous competition) developed different and new approaches that helped him achieve the first place in this competition. Based on this we can conclude that adaptations of the previous model did not outperform the top ranking models in this competition. In this full manuscript we now also showed finer class activation maps that focus on subcellular regions from the single cell classification model (this competition) compared to the image classification model (previous competition), as well as use the single-cell classification model to inspect heterogeneity in single cell labels among images with multiple labels in the published HPAv20.

The problem is not how to simultaneously segment and assign labels to single cells, the problem is how to assign labels to pre-segmented cells when they have not been annotated individually. Framing the challenge in this way would bring interesting methods and would help answer biological questions related to protein localization patterns that have not been studied before at single cell resolution.

By providing a baseline segmentation model the competition is close to the framing as phrased by the reviewer. As described above we made the choice to keep an open competition design to encourage diversity and innovation in solutions, and will make sure to clarify this in the full manuscript. We thank the reviewer for seeing the value of this challenge in terms of novel interesting methods and possibilities to study previously intractable biological questions.

Reviewer #2:**Remarks to the Author:**

This paper will describe the results of an ongoing Kaggle competition, which aims at generating single cell level protein localization labels from training set annotated at the image level, in order to place the competitors in a weakly-supervised setting, alleviating the burden of obtaining hand-labeled data sets. The weakly-supervised aspect of this multi-label classification problem relies on 1) annotation not directly giving the expected solution for this task (imprecise label), because the class are given at the full image level and with no single cell identification and 2) the use of existing resources (previous competition ref [14]).

Compared to the first competition which was at the image level, the purpose is to target single-cell variations with task being to perform segmentation (of cells only, see below?) and classification. Another main difference is that the code to apply a model will all be run from jupyter notebooks, allowing a fair assessment of code performance and giving constraint to the application of the models, which will be trained independantly. The challenge is a big add on to the previous HPA image challenge, now focusing on single cells, and the analysis of results is expected to follow the same lines as the first paper (ref 14) with some adjustments regarding cell segmentation which may be inherent to some solutions, and new interpretation of the atlas allowed by the developed methods which will allow to unprecedentedly augment the quantity of available information in the cell atlas important resource, and open exciting possibilities when shared. The authors proposed for example to explore the applicability of the trained models to study cell cycle dependant variations which will be indeed of great interest. The analysis of results will give the performance of pattern class identification, but also of the auxiliary task of cell mask segmentation. It will also try to analyse in depth the different best performing models with visual maps of class activation CAM (only for cnn-based solutions) and visualisation of subcellular clusters. An ablation study is also aimed to be performed.

We thank the reviewer for his/her constructive comments, and appreciate that he/she sees that this is a big add-on to the previous competition and sees both the need and benefit for the scientific community of the development of such models.

Because of the auxiliary task of cell segmentation, and of the introduction enhancing the importance of quantification of expression in addition to their spatial patterns, I found that a clear introduction was missing regarding the expected task.

We thank the reviewer for pointing this out. To clarify the task of the competition we have modified the introduction and included a figure (Figure 2 below) to the full manuscript.

It is still unclear for me, in particular regarding the introduction also emphasizing the fact that one protein can have several localization in one cell and also that these localization can be unbalanced, if the participants task was to either identify single cells, and give the list of localization label (with no position in the image apart) to the protein of interest OR to identify single cells, and to perform instance segmentation of proteins (i.e pixel segmentation and spatial patterns class). This may come from the lack of figures at this stage, but some text edition could also help to make it clearer.

We agree with the reviewer that a schematic figure would immensely help to clarify this point and we have included Figure 2A,B (see below) in the full manuscript. In this competition, the participants are asked to identify single cells and give all localization labels for every cell.

The main challenge of this competition is single cell classification on a weakly labeled training dataset (Figure 2A). The segmentation of each cell is required to identify which cell corresponds to which label(s). The participants had the possibility (but were not required) to make segmentation as an auxiliary task to the classification model, which could potentially improve both the segmentation and classification results, and make the solution more generalizable by understanding the context from which the labels are predicted. Instead of closing the competition in a mode with segmentation followed by classification, **we deliberately chose this open design to allow participants the flexibility to design their solution to solve both problems simultaneously or separately.** This open design allows participants to be creative in their approaches (Figure 2B).

Figure 2. Challenge overview. **A.** Dataset setup for the competition. For training, participants were provided access to a well balanced training set and the public HPA dataset, which consists of fluorescent images with four channels and corresponding image-level labels. The solutions' performance were assessed by a private dataset of 1776 images containing 41k single cells with corresponding single-cell labels. This dataset was divided into the public testset, which was used by the participants to test their models for a ranking on the live leaderboard, and the private testset, in which all images and their single-cell labels were completely hidden and used for the final ranking. **B.** Overview of main approaches: image-level models take in images and predict image-level labels, which can be combined with class activation maps of different classes and segmented cell regions to give the final cell-level labels. Image-level models can also be trained on whole images, but predict on bag-of-cell tiles at inference to get cell labels. Cell-level models take in single cell crops and predict single cell labels.

It is mentioned that single-cells level label with all labels corresponding to image-labels were removed, I could not figure out why by myself, could you elaborate? Also in your previous paper [14] it was stated : "However, we expect the impact to be minimal because only ~2% of proteins vary in their localization patterns between cells in images" : then how many proteins were selected following this rule of imposing image with cells showing variability?

There are several differences between the last competition and this one. One of the most important differences is that in the last challenge, we focused on aggregated labels of all cells in images, so we purposely removed most images with high single-cell variation (only 2% of the remaining images had single cell variations in this dataset). While in this challenge, we aim to study the heterogeneity of cells and proteins in the same images, therefore the test set was purposely chosen to have high single-cell variation within images. In total, about 20% of human proteins mapped in HPA show pronounced single cell variability (Mahdessian *et al.* "Spatiotemporal dissection of the cell cycle with single-cell proteogenomics". Nature 590, 649–654 (2021). <https://doi.org/10.1038/s41586-021-03232-9>). The confusion is understandable, and we have clarified this in the manuscript now, including the sentence "*To evaluate the participating models' performance, a dataset of 41,597 cells in images with pronounced single cell heterogeneity were manually annotated for single-cell labels...*"

Because this challenge is built upon the previous one [ref 14] , with annotations and training data sets still available, it would be of interest to create a figure that would clearly compare the task in both competitions, and the training data set level of annotations. In particular because the previous dataset is released and made available to the participants to train their model, it is important to underline how different is the task so it can still be considered as weakly supervised, in particular because most of the image from the first competition was presenting uniform cell class (meaning one type of localization/label) , so it could be used directly as additional training data set by simply segmenting the cells, so the challenge here would have been not that weakly supervised. Could you comment on this point to justify the usage of weakly supervised?

We thank the reviewer for this great suggestion. We included Figure 2A to clarify the task of this competition and explanatory text describing the datasets and how the participants could make use of "external" data sets such as from HPA or the previous competition (Figure 2A, Methods). In the full manuscript we have included a supplementary Table where we compare all aspects of the two competitions, including tasks, labels, datasets and metrics (Supplementary Table 4).

In the first challenge, the majority of training images were picked to present uniform cell classes (i.e. avoid single cell heterogeneity) because the task was focused on image-level classification. In this challenge, the data set (particularly the test sets) was sampled to have high single cell heterogeneity,

which means that the individual cells will not inherit exactly the same classes as image-labels. These make our training set a weakly labeled dataset. To score high on the test set, the model needs to handle these image-level weak labels and predict cell-level labels, particularly high cell-to-cell variability in spatial localisation. The naive solution (image-classification) might work for classes with many cell instances (e.g. Nucleoplasm) but it is unlikely to work for rare and high heterogeneous classes like Mitotic Spindle. For rare classes, you can't get enough training instances after the above mentioned exclusion. In fact, after the competition we indeed saw our challenge design stimulated a large variety of solutions, none of the winning solutions simply train models only on segmented cells with image-level labels.

The dataset represents 17 different cell lines, and 19 class including the negative one are considered, while the first challenge considered 28 labels and 27 different cell lines. The explanation about the reduction of class (getting a simplified balanced dataset) comes very late in the method, but there is no explanations about the reduction in cell lines. Please add a few words about this choice in the presentation of the challenge.

We thank the reviewer for noticing this detail and will clarify this in the manuscript. Even though we have manually annotated over 41,000 single cells, this dataset is much smaller than the whole HPA (~1.5 million cells). Given the emphasis on single cell annotation of images with pronounced heterogeneity, there is an even greater imbalance for both cell lines and classes in the single-cell labeled dataset. So the decision to merge/exclude was to have a simplified and more balanced dataset in terms of both cell lines and classes. We have added the following text to the Methods explaining the reduction in cell lines: *"The final test set consisted of 1,776 images comprising 41,597 single cells from 17 human cell lines of varying morphology.... To match the test set, the training set was constructed by sampling images from the 17 cell lines above from the training set of the previous Kaggle challenge.... Furthermore, all publicly available HPA images of all cell lines were available as extra training data."*

Considering the difficulty of this challenge, we also decided to remove "difficult" cell lines where artifacts are common.

In particular in the methods it is specified that 30 labels were used for the annotation process, and the grouping in 19 class is summarized as "functionally and spatially similar", giving the list of grouped class. For example Focal adhesions (FAs) and Actin filaments has been grouped as Actin Filament: this is for example a choice that could be discussed because FAs have usually a very different shape than filament, and I do not think they can be considered to have the same function.

We agree with the reviewer that FAs and Actin filaments are indeed functionally different. The FA class was too small to include in itself, and since oftentimes proteins are localized to both FAs and actin filaments, and the fact that FAs are found at the tip of Actin filaments we reasoned that this was the most reasonable merger. We have updated the manuscript text to *"To simplify the challenge and*

balance the class distribution, we prioritized and grouped classes that are functionally and spatially similar into 19 classes...”

Ablation studies are of interest to try to understand better the causality of the obtained results, however in the previous study [14] the results were underexploited. Maybe focusing the ablation study on the input data rather than changing the architecture of the loss function could lead to a better interpretation of the robustness of the different models and their applicability for the readers. We agree with the reviewer that ablation on input is helpful to test the robustness of models. The winning teams in our competition found that augmentations of inputs at training and inference are important (Supplementary Notes 1-4), so there is no doubt that it contributed to the robustness of the models. We have also discussed other ablation experiments and their results in the Supplementary Notes.

Super minor:

The proportion /list of labels was not given (to detect "rare" class).

Correct, the proportion of labels were purposely kept hidden because the competition was ongoing at that time, and we did not want to risk any information leakage to competition participants. We have now included the portion/list of labels for training data and test dataset at Figure 2C, Supplementary Table 1A.

While domain knowledge is interesting to be assessed among the participants, it is not clear what is the definition of domain here, and this should be defined in the question asked to the participants. In the post-competition survey, we defined domain knowledge as “knowledge in biology or microscopy imaging”. After closing the competition and conducting the survey, we now know that 2 or the top 4 teams had domain knowledge in this competition, in contrast to in the previous competition where none of the top 5 teams did. This supports our assumption that this was a hard competition, where experienced participants and/or domain-knowledge experts had to team up to achieve high ranks.

Reviewer #3:

Remarks to the Author:

The authors propose a competition to evaluate cell segmentation and classification methods for the purpose of detecting, localizing, and identifying protein patterns in fluorescence microscopy images displaying multiple cells. The competition is open to the community and aims to spur the development of novel segmentation and classification approaches capable of dealing with the limited availability of annotations at the single-cell level.

This competition is timely, as single-cell analyses in imaging (as well as in other technologies, such as scRNA-seq technologies) become prevalent. The classification task is certainly challenging in that methods will have to deconvolve multiple image-level labels onto high resolution single-cell predictions. The fact that the competition will allow the use of additional sources of data beyond those provided by the organizers is aligned with current trends in machine learning positing that both data and methods are key for performance breakthroughs.

We thank the reviewer for appreciating the challenge's motivation and the promise of single cell image classification models.

One comment is related to the fact that cell phenotypes are notoriously hard to bin onto well defined categories, mostly because one observes a continuum of phenotypes as opposed to discrete patterns (Liberali, P., Snijder, B. & Pelkmans, L. Single-cell and multivariate approaches in genetic perturbation screens. *Nat Rev Genet* 16, 18–32 (2015)). While a classification task certainly makes it easier to evaluate different methods, the authors are encouraged to provide a discussion on other end-points for evaluation (e.g., phenotypic distances among protein pattern populations) that would better translate to the novelty-seeking scenarios where most biology studies are undertaken.

It is true that cell phenotypes are often continuous, particularly when studying perturbed cells (which is not the case in this challenge). The data we have in the competition are cell lines in normal culture condition (log phase growth) and the cell phenotypes will for example represent a continuum of cell cycle states. With regard to the protein localization patterns, they can most often be categorized into well defined classes due the organellar compartmentalization of the cell (Thul *et al.* "A subcellular map of the human proteome". *Science* 356, eaal3321 (2017). <https://doi.org/10.1126/science.aal3321>). Nevertheless, due to the continuous transitions of cell states and protein localized to multiple compartments, the boundary between different patterns can be blurred.

We agree with the reviewer that the models developed in this competition can be proven useful in novelty-seeking scenarios. Even though the models have been trained on discrete labels, it is still possible to obtain a continuous representation of the localization patterns in feature space. In fact, by using UMAP visualization to show the learned features of the winning model from our last image-level classification competition (Ouyang *et al.* "Analysis of the Human Protein Atlas Image Classification competition". *Nat Methods* 16, 1254–1261 (2019). <https://doi.org/10.1038/s41592-019-0658-6>), we manage to identify nucleolar protein clusters corresponding to a novel dynamic subcompartment lining the rim of the nucleolus (Stenström *et al.* "Mapping the nucleolar proteome reveals a spatiotemporal organization related to intrinsic protein disorder". *Mol Syst Biol.* Aug;16(8):e9469 (2020). doi: 10.15252/msb.20209469. PMID: 32744794; PMCID: PMC7397901). More importantly, these image-level features were used, in combination with protein–protein associations AP-MS data, to discover several

dozens of undocumented subcellular systems (Qin *et al.*, “A multi-scale map of cell structure fusing protein images and interactions”. *Nature* 600, 536–542 (2021). <https://doi.org/10.1038/s41586-021-04115-9>). We envisioned the features extracted from single-cell models will be much finer and exact. Using the above and other multi-omics integration approaches, we and others will explore the HPA Cell Atlas for novel patterns at the single cell level in subsequent follow up studies, including analysis of subcellular patterns - phenotype continuum (ongoing).

The evaluation criterion for the proposed study merges the performance of both segmentation and classification tasks. This is fine but it would be good to have in addition more details on the segmentation task alone. In that regard, there is already an on-going cell segmentation benchmarking effort (cf. <http://celltrackingchallenge.net/latest-csb-results/>; see also Ulman, V., Maška, M., Magnusson, K. *et al.* An objective comparison of cell-tracking algorithms. *Nat Methods* 14, 1141–1152 (2017)) to which the current study should better align in terms of terminology (e.g., the IoU metric is the Jaccard index in the cell segmentation benchmark) as well as metrics and ranking approach.

We agree with the reviewer that the segmentation can be approached separately, but see two key arguments for the weakly-supervised challenge setting. First, even with well-segmented cell masks, obtaining the amount of ground-truth labels for individual cells for fully supervised models is too costly to implement in practice. Second, providing a weakly-labeled dataset will help to draw attention from experts from the AI field and use it as a benchmark dataset for development of new weakly-supervised algorithms. Therefore we decided to design the challenge to encourage innovation in making use of image-level datasets for precise cell level classification.

Another practicality of the chosen evaluation metrics stem from the fact that Kaggle platform does (unfortunately) not support double metrics. Therefore, we chose to use mAP, which is a popular metric for similar computer vision tasks, as classification metric (our main task) for all cells that passed the 0.6 IOU threshold. We noted in the full manuscript “*Since most teams made use of the HPACellSegmentator (Supplementary Table 2), with several community-developed improvements such as faster runtime, no significant correlation can be seen between segmentation score and AP score (Supplementary Figure 1).*”

Another issue is related to the fact that images where all cells have the same label as the label associated with the entire image are removed (i.e., images with entirely homogeneous populations). These images are actually a baseline case that the methods should be able to handle as well, so it's not clear why these images are being removed. In general, it's not too clear to me why the study is focusing only on pronounced heterogeneity.

The test set is chosen to be of pronounced heterogeneity to ensure that a model needs to figure out the exact label for each cell to be successful, while models naively trained on segmented cells with image-level labels will give baseline results are unlikely to succeed in this challenge setting. In fact, now when the competition is closed we can see that our challenge design stimulated a large variety of solutions,

where none of the winning solutions simply train models only on segmented cells with image-level labels (See full Supplementary Table 2).

We assume that if the model can handle high heterogeneity images well then it will also be able to handle a more homogenous population. We used inceptionv3 - the best scoring single model from Team 1 - to predict all single cell labels in HPAv20, which subsequently gave some interesting biological insights about the nature of cell heterogeneity (See section *Exploring Single Cell Variability in the HPA Subcellular Atlas*).

Team	Rare label strategies		Segmentation strategies		Single cell strategies		Loss strategies	
	Pseudo-labeling	Manual labeling of rare class	Border cell classifier	Edge heuristics	Transformers	Cell and image models	Weighed -loss / Oversampling	Combine 2+ losses
Team 1	X		X	X	X	X		X
Team 2			X	X		X	X	
Team 3	X	X				X	X	X
Team 4	X	X		X				
Team 5	X	X						
Team 6 *								
Team 7	X			X		X		
Team 8				X				
Team 9						X		
Team 10				X		X		

Definitions	
Pseudo-labeling:	Predict labels for unlabeled data to increase the size of the training set.
Manual labeling:	Manually label images containing rare classes, to increase the size of the training set.
Border cell classifier:	Separate classifier to remove non-whole cells from input images.

Edge heuristics:	Add heuristics to remove cells touching edges from input images.
Transformers:	The use of a transformer model for cell prediction
Cell and image models:	The usage of separate models for image and cell predictions.
Weighted loss / oversampling:	Using either weighted loss values or oversampling strategies for more efficient loss values.
Combine 2+ losses:	Using at least 2 different loss functions.

Table 1: Overview of different approaches used by the top 10 teams in the competition. *Team 6 did not use or did not supply information about these methods in their overview. For full summary, please refer to Supplementary Table 2.

A more specific question relates to the private and public test sets. It would be good to disclose (either before or after the competition) the degree of overlap (in terms of image or phenotypic similarity) between them.

We assume the reviewer did not mean “overlap” but instead “similarity” between images. There is **no overlap** between private and public test sets. Instead, they are similarly distributed in terms of labels and cell lines. This is crucial to the success of challenges like this where participants rely on the public leaderboard for validation (unfortunately sometimes neglected, hence leading to major leaderboard shake-ups such as in recent bioimage competitions <https://www.kaggle.com/c/hubmap-kidney-segmentation/leaderboard> or <https://www.kaggle.com/c/vinbigdata-chest-xray-abnormalities-detection/leaderboard>). We were extremely careful when designing the challenge to ensure similar distributions in the different data sets, and the success can now be confirmed by the minimal shake-up in the final leaderboard (<https://www.kaggle.com/c/hpa-single-cell-image-classification/leaderboard>). We supplied Supplementary Table 1, together with Figure 2C with all details about the distribution of public and private test sets in the full manuscript.

One suggestion for the analysis is to investigate whether cell-to-cell interactions (c.f. Bechtel, T.J., Reyes-Robles, T., Fadeyi, O.O. et al. Strategies for monitoring cell–cell interactions. *Nat Chem Biol* 17, 641–652 (2021)) or agglomerations are informative with regards to heterogeneity and pattern distributions as well as to method performance. This should be possible to analyze as single-cell locations and outlines would be provided by the methods.

We agree with the reviewer that there are many interesting aspects of cell-to-cell variability that can be explored given these models, one being which proteins change their location in relation to cell-cell interactions or cell crowding. Although these biological questions remain out of the scope of this Analysis Article, these are the types of questions that can be now addressed in subsequent studies as the single-cell classification model is available.

Decision Letter, second revision:

Dear Emma,

Thank you for submitting your revised manuscript "Analysis of the Human Protein Atlas Weakly-Supervised Single Cell Classification Competition" (NMETH-AS44310C). It has now been seen by the original referees and their comments are below. The reviewers find that the paper has improved in revision, and therefore we'll be happy in principle to publish it in Nature Methods, pending minor revisions to satisfy the referees' final requests and to comply with our editorial and formatting guidelines.

For revisions, we ask that you address their requests for changes to the text, and also that you tone down or clarify claims of performance, as raised by ref 2. In this case, you may point to any deficiencies that remain in the overall performance (at least for certain classes) as areas that need continued study.

TRANSPARENT PEER REVIEW

Nature Methods offers a transparent peer review option for new original research manuscripts submitted from 17th February 2021. We encourage increased transparency in peer review by publishing the reviewer comments, author rebuttal letters and editorial decision letters if the authors agree. Such peer review material is made available as a supplementary peer review file. Please state in the cover letter 'I wish to participate in transparent peer review' if you want to opt in, or 'I do not wish to participate in transparent peer review' if you don't. Failure to state your preference will result in delays in accepting your manuscript for publication.

Thank you again for your interest in Nature Methods Please do not hesitate to contact me if you have any questions.

Sincerely,
Rita

Rita Strack, Ph.D.
Senior Editor
Nature Methods

ORCID

Reviewer #1 (Remarks to the Author):

The authors have addressed all of my concerns and have presented a greatly improved manuscript. I want to thank them for clarifying these points extensively. It is worth noting that the open design of the competition targeted a very challenging problem with a formulation that stimulated diverse strategies and solutions. Similarly, the careful organization of the challenge has resulted in important progress in the state-of-the-art. I am very pleased with the study presented in this manuscript and congratulate the authors for their efforts to conduct this piece of research, and to disseminate their findings.

The following are minor suggestions to improve the content of this manuscript:

Line 79: Mask RCNN is a fully supervised model requiring segmentation masks for segmentation and bounding boxes for object detection. I understand that the authors refer to more specific adaptations for weakly supervised learning in medical imaging, so clarifying that in the main text (as they did in the responses), may help the reader understand and avoid confusions.

The discussion suggests that the mAP metric does not use thresholds to estimate performance (in contrast to F1), but the methods section and other parts of the paper indicate that the chosen IoU threshold was 0.6. Also, in the mAP definition, is it the case that detections are ranked by classification score?

Proofread the text for typos, such as: “single cell something” should be connected with the hyphen: “single-cell something”. Examples: line 94, 96, 106, ... Other types of typos were observed but not documented.

The descriptions of the top participants also need proofreading and editing. It is recommended to paraphrase their descriptions to make them more accessible and useful for future researchers to follow their ideas.

Formatting of tables and figures can be improved in the supplementary materials.

Line 201 reads "All the solutions in this competition were based on CNNs" but seems like transformers involved too. Perhaps the authors mean deep neural networks, more generally speaking.

In some parts of the text, the authors write in future tense, when the analysis has been performed already: Line 431 Grad-CAM will be used. Line 478 some ablations will be performed.

Reviewer #2 (Remarks to the Author):

This paper describe the results of a Kaggle competition, which aims at generating single cell level protein localization labels from training set annotated at the image level, in order to place the competitors in a weakly-supervised setting, alleviating the burden of obtaining hand-labeled data sets.

The weakly-supervised aspect of this multi-label classification problem relies on 1) annotation not directly giving the expected solution for this task (imprecise label), because the class are given at the full image level and with no single cell identification and 2) the use of existing resources (previous competition HPA for a different task).

The challenge is a big add on to the previous HPA image challenge, now focusing on single cells.

Compared to the previously reviewed registered report, which was not yet presenting the results, the manuscript is much more clearer and have adressed all the points I had raised.

However now that results are presented, a demonstration about the exploitability of the results and next steps of exploitations is missing:

- only 6 over 19 classes have a mean score above 0.5, meaning that the classification failed in most cases for the other classes.

- The authors stated "We demonstrated that the learned feature embedding produced by the winning models can shed light to single cell spatial variability within images, protein expression heterogeneity across cell lines, cell cycle and potentially performing different functions in different organelles.". This is actually discussed and presented in supplementary figure 9, but does not really or clearly directly use the results from the winning team. While it is suggested, I think this is overstated since the study does not take into account the results which were low for some class, and about 0.6 for the best. According to

me the exploitability of results with the obtained level of accuracy obtained by the winning teams is lacking, and in addition is not discussed.

Minor comments:

Figure 3B: MitonNdria Typo twice on the figure

Figure 2D: (up to the editor but maybe rather but the numbers of violin plot in a sup table?)

Author Rebuttal, second revision:

RESPONSE TO REVIEWER COMMENTS

Reviewer #1 (Remarks to the Author):

The authors have addressed all of my concerns and have presented a greatly improved manuscript. I want to thank them for clarifying these points extensively. It is worth noting that the open design of the competition targeted a very challenging problem with a formulation that stimulated diverse strategies and solutions. Similarly, the careful organization of the challenge has resulted in important progress in the state-of-the-art. I am very pleased with the study presented in this manuscript and congratulate the authors for their efforts to conduct this piece of research, and to disseminate their findings.

We thank the reviewer for their acknowledgement of the challenging problem this competition was trying to solve and our efforts in the challenge organization. We hope that the results of this competition can push the state-of-the-art and aid new discoveries.

The following are minor suggestions to improve the content of this manuscript:

Line 79: Mask RCNN is a fully supervised model requiring segmentation masks for segmentation and bounding boxes for object detection. I understand that the authors refer to more specific adaptations for weakly supervised learning in medical imaging, so clarifying that in the main text (as they did in the responses), may help the reader understand and avoid confusions.

Thanks for pointing this out, we have now clarified this in the manuscript.

“For example, Class Activation Maps (CAM)-guided approaches and weakly-supervised adaptations of mask RCNN were designed to localize signal regions and assign precise object labels given coarse image-level labels.”

The discussion suggests that the mAP metric does not use thresholds to estimate performance (in contrast to F1), but the methods section and other parts of the paper indicate that the chosen IoU threshold was 0.6. Also, in the mAP definition, is it the case that detections are ranked by classification score?

We would like to clarify that there is indeed no threshold used for mAP itself but the threshold was used for punishing poor segmentation results before calculating mAP. More specifically, mAP scoring can be

seen as a 2-stage process, and that the AP score is calculated based on a selected IoU threshold (in our case 0.6; i.e. at least 60% cell segmentation match), thereby usually referred to as mAP_{0.6} or mAP@0.6IoU. The first step is to find all the cells that matched with ground-truth at IoU of 0.6. The second step continues with mapped, unmapped ground-truth cells and unmapped predicted cells as TP, FN and FP respectively. At this stage, for each class, predictions are ranked according to their confidence score, and precision and recall are calculated at different confidence thresholds. AP is the area under the precision/recall curve. In this sense, AP calculation does not depend on the individual class threshold. So the short answer is yes mAP depends on the threshold of IoU, in contrast to F1 which depends on the threshold of confident score of each class.

Proofread the text for typos, such as: “single cell something” should be connected with the hyphen: “single-cell something”. Examples: line 94, 96, 106, ... Other types of typos were observed but not documented.

We have proof-read the article and updated these typos.

The descriptions of the top participants also need proofreading and editing. It is recommended to paraphrase their descriptions to make them more accessible and useful for future researchers to follow their ideas.

We have worked over the method descriptions of the 4 winning teams to be written more in the same language to increase reproducibility.

Formatting of tables and figures can be improved in the supplementary materials.

Line 201 reads “All the solutions in this competition were based on CNNs” but seems like transformers involved too. Perhaps the authors mean deep neural networks, more generally speaking.

We thank the reviewer for noticing this and have updated the wording accordingly.

In some parts of the text, the authors write in future tense, when the analysis has been performed already: Line 431 Grad-CAM will be used. Line 478 some ablations will be performed.

We thank the reviewer for noticing these errors stemming from registered report stage 1. We have updated the tense of the analysis performed.

Reviewer #2 (Remarks to the Author):

This paper describes the results of a Kaggle competition, which aims at generating single cell level protein localization labels from training set annotated at the image level, in order to place the competitors in a weakly-supervised setting, alleviating the burden of obtaining hand-labeled data sets.

The weakly-supervised aspect of this multi-label classification problem relies on 1) annotation not directly giving the expected solution for this task (imprecise label), because the class are given at the full

image level and with no single cell identification and 2) the use of existing resources (previous competition HPA for a different task).

The challenge is a big add on to the previous HPA image challenge, now focusing on single cells.

We thank the reviewer for their constructive comments, and appreciate that they see this as a big add-on to the previous competition and acknowledge the need for and benefit of the development of such models for the greater scientific community

Compared to the previously reviewed registered report, which was not yet presenting the results, the manuscript is much more clearer and have addressed all the points I had raised.

However now that results are presented, a demonstration about the exploitability of the results and next steps of exploitations is missing:

- only 6 over 19 classes have a mean score above 0.5, meaning that the classification failed in most cases for the other classes.

We would like to clarify that the winning model with which we choose to do subsequent analysis has a performance range of AP 0.4-0.8 (13/19 above 0.5), corresponding to a mAP@0.6IOU of 0.57 for the entire testset. The classes that this model (or overall for other models) performed worse at are merged classes (eg, Vesicles & Punctate Patterns, Centrosome) or classes with non-distinctive patterns even for humans (ER, Negative, Nuclear Bodies). Please see our answer to the comment regarding scores above, the models do by no means fail in most cases for the other classes. The model is of course not perfect, but we believe it is good enough to reveal novel insights to subcellular biology, as the previous winning image classification model has been. To avoid confusion, we have added a section into the discussion: *“As with any other machine learning work, our winning models carry the bias of the training data, and the performance on some classes are better than others. Although the best single model performance reaches AP 0.6 for half of the classes, even 0.8 for some classes (Supplementary Table 3), rarer classes have a considerable gap from human expert performance (accuracy ~0.9 based on agreement among multi-annotators). Developments in self-supervised, unsupervised and few-shot learning could potentially tackle high class imbalance, rare class or novelty detection to a much greater extent...”*

It is also worth noticing that our detection threshold or segmentation matching threshold (0.6) is higher than other object detection competitions. AP score is calculated based on selected IoU threshold (in our case 0.6, or at least 60% bounding box match), thereby usually referred to as mAP60 or mAP@0.6IOU. IOU threshold can greatly influence the mAP scores (lower IOU usually boosts AP thereby mAP score). Often, 0.5 is used as IOU threshold, such as in the famous Open Images Detection challenge (https://storage.googleapis.com/openimages/web/evaluation.html#instance_segmentation_eval). Another example is another Kaggle competition challenge that used mAP like our competition and was hosted in the same year, VinBigdata Chest X-ray used mAP@0.4IOU.

- The authors stated “We demonstrated that the learned feature embedding produced by the winning models can shed light to single cell spatial variability within images, protein expression heterogeneity across cell lines, cell cycle and potentially performing different functions in different organelles.”. This is actually discussed and presented in supplementary figure 9, but does not really or clearly directly use the results from the winning team. While it is suggested, I think this is overstated since the study does not take into account the results which were low for some class, and about 0.6 for the best. According to me the exploitability of results with the obtained level of accuracy obtained by the winning teams is lacking, and in addition is not discussed.

We thank the reviewer for his/her constructive comments, and would like to clarify that Supplementary Figure 9 is actually using the result from the best model of the winning team, as discussed in Methods.

We have updated this sentence to reflect this fact more clearly:

“For proteins with multi-locations on image labels, but only showed single location at the single-cell level as discovered from the result of inceptionv3 model in the previous section - Single-cell prediction, gene set enrichment analysis (GSEA) was performed....”

Please see our comments on how to interpret the score above. As mentioned previously, we have also added text in the discussion about limitations of the models and future direction, specifically:

“As with any other machine learning work, our winning models carry the bias of the training data, and the performance on some classes are better than others.... Developments in self-supervised³⁷, unsupervised³⁸ and few-shot³⁹ learning could potentially tackle high class imbalance, rare class or novelty detection to a much greater extent. ”

Minor comments:

Figure 3B: MitonNdria Typo twice on the figure

Figure 2D: (up to the editor but maybe rather but the numbers of violin plot in a sup table?)

We thank the reviewer for pointing out our typos. We have fixed these and moved the descriptions with numbers of the Violin plot to Supplementary Figure 3.

Final Decision Letter:

Dear Emma,

I am pleased to inform you that your Analysis, "Analysis of the Human Protein Atlas Weakly-Supervised Single Cell Classification Competition", has now been accepted for publication in Nature Methods. Your paper is tentatively scheduled for publication in our October print issue, and will be published online prior to that. The received and accepted dates will be April 14, 2021 and August 10, 2022. This note is intended to let you know what to expect from us over the next month or so, and to let you know where to address any further questions.

Please note that Nature Methods is a Transformative Journal (TJ). Authors may publish their research with us through the traditional subscription access route or make their paper immediately open access through payment of an article-processing charge (APC). Authors will not be required to make a final decision about access to their article until it has been accepted. Find out more about Transformative Journals

Authors may need to take specific actions to achieve compliance with funder and institutional open access mandates. If your research is supported by a funder that requires immediate open access (e.g. according to Plan S principles) then you should select the gold OA route, and we will direct you to the compliant route where possible. For authors selecting the subscription publication route, the journal's standard licensing terms will need to be accepted, including self-archiving policies. Those licensing terms will supersede any other terms that the author or any third party may assert apply to any version of the manuscript.

Your paper will now be copyedited to ensure that it conforms to Nature Methods style. Once proofs are generated, they will be sent to you electronically and you will be asked to send a corrected version within 24 hours. It is extremely important that you let us know now whether you will be difficult to contact over the next month. If this is the case, we ask that you send us the contact information (email, phone and fax) of someone who will be able to check the proofs and deal with any last-minute problems.

If, when you receive your proof, you cannot meet the deadline, please inform us at rjsproduction@springernature.com immediately.

Once your manuscript is typeset and you have completed the appropriate grant of rights, you will receive a link to your electronic proof via email with a request to make any corrections within 48 hours. If, when you receive your proof, you cannot meet this deadline, please inform us at rjsproduction@springernature.com immediately.

Once your paper has been scheduled for online publication, the Nature press office will be in touch to confirm the details.

Once your paper has been scheduled for online publication, the Nature press office will be in touch to confirm the details.

Content is published online weekly on Mondays and Thursdays, and the embargo is set at 16:00 London time (GMT)/11:00 am US Eastern time (EST) on the day of publication. If you need to know the exact publication date or when the news embargo will be lifted, please contact our press office after you have submitted your proof corrections. Now is the time to inform your Public Relations or Press Office about your paper, as they might be interested in promoting its publication. This will allow them time to prepare an accurate and satisfactory press release. Include your manuscript tracking number NMETH-AS44310D and the name of the journal, which they will need when they contact our office.

About one week before your paper is published online, we shall be distributing a press release to news organizations worldwide, which may include details of your work. We are happy for your institution or funding agency to prepare its own press release, but it must mention the embargo date and Nature Methods. Our Press Office will contact you closer to the time of publication, but if you or your Press Office have any inquiries in the meantime, please contact press@nature.com.

Nature Portfolio journals encourage authors to share their step-by-step experimental protocols on a protocol sharing platform of their choice. Nature Portfolio 's Protocol Exchange is a free-to-use and open resource for protocols; protocols deposited in Protocol Exchange are citable and can be linked from the published article. More details can found at www.nature.com/protocolexchange/about.

Best regards,
Rita

Rita Strack, Ph.D.
Senior Editor
Nature Methods